# Statistical emulation of a perturbed basal melt ensemble of an ice sheet model to better quantify Antarctic sea level rise uncertainties

Mira Berdahl[1,3], Gunter Leguy[2], William H. Lipscomb[2], and Nathan M. Urban[3,4]

[1]Department of Earth and Space Sciences, University of Washington, WA, USA
[2]Climate and Global Dynamics Laboratory, National Center for Atmospheric Research, Boulder, CO, USA
[3]Computational Physics and Methods Group, Los Alamos National Laboratory, Los Alamos, NM, USA
[4]Computational Science Initiative, Brookhaven National Laboratory, Upton, NY, USA

**Correspondence:** Mira Berdahl (mberdahl@uw.edu)

**Abstract.** Antarctic ice shelves are vulnerable to warming ocean temperatures, and some have already begun thinning in response to increased basal melt rates. Sea level is therefore expected to rise due to Antarctic contributions, but uncertainties in its amount and timing remain largely unquantified. In particular, there is substantial uncertainty in future basal melt rates arising from multi-model differences in thermal forcing and how melt rates depend on that thermal forcing. To facilitate uncertainty quantification in sea level rise projections, we build, validate, and demonstrate projections from a computationally efficient statistical emulator of a high resolution (4 km) Antarctic ice sheet model, the Community Ice Sheet Model version 2.1. The emulator is trained to a large (500-member) ensemble of 200-year-long 4-km resolution transient ice sheet simulations, whereby regional basal melt rates are perturbed by idealized (yet physically informed) trajectories. The main advantage of our emulation approach is that by sampling a wide range of possible basal melt trajectories, the emulator can be used to (1) produce probabilistic sea level rise projections over much larger Monte Carlo ensembles than are possible by direct numerical simulation alone, thereby providing better statistical characterization of uncertainties, and (2) predict the simulated ice sheet response under differing assumptions about basal melt characteristics as new oceanographic studies are published, without having to run additional numerical ice sheet simulations. As a proof-of-concept, we propagate uncertainties about future basal melt rate trajectories, derived from regional ocean models, to generate probabilistic sea level rise estimates for 100 and 200 years into the future.

## 1 Introduction

### 1.1 The physical origin of Antarctic sea level rise uncertainties

Mass loss from Antarctica over the past several decades has primarily been a result of melt at the base of ice shelves (Cook et al., 2016; Rintoul et al., 2016; Pritchard et al., 2012; Rignot and Jacobs, 2002). Depoorter et al. (2013) found that about

half of the ice-sheet surface mass gain is lost through oceanic erosion before reaching the ice front. Basal melt weakens the backforce on upstream glaciers which causes grounding line retreat (Konrad et al., 2018; Rignot et al., 2014), increases flow rate (Pattyn, 2018), depresses surface heights of grounded ice (Konrad et al., 2017), and ultimately impacts sea level. Antarctic ice loss is particularly susceptible to a positive feedback due to the so-called marine ice shelf instability (MISI) (Weertman, 1974; Schoof, 2007). Much of West Antarctica's ice is grounded below sea level, with a retrograde bed sloping downward toward the interior of the continent. MISI theory suggests that increased basal melt rates beneath some key WAIS ice shelves (e.g., Pine Island and Thwaites) could result in an unstable grounding line retreat causing runaway ice loss for the entire region. In fact, there is some evidence through observations and modeling that this process may have already been triggered (Rignot et al., 2014; Joughin et al., 2014; Favier et al., 2014). Forcing due to basal melt is therefore likely to become an increasingly dominant contributor to Antarctic SLR (Bulthuis et al., 2019). Despite its potential to contribute to sea level rise (SLR) vastly more than any other single source ($\sim 5$ m West Antarctica, $\sim 60$ m all Antarctica), and documented ice shelf thinning (e.g. Schroeder et al., 2019; Reese et al., 2018), Antarctica's contribution to future sea level remains highly uncertain (Oppenheimer et al., 2019; Heal and Millner, 2014).

The primary unknown in how basal melting will affect sea level rise is the uncertainty in sub-shelf melt rates themselves. Future basal melting is uncertain because it depends on unresolved, coupled ice-ocean processes which are in turn driven by a range of global and regional ocean and atmospheric conditions. Sub-shelf melting can be decomposed into several factors including: changes to the large scale circulation in the Southern Ocean and cross-slope exchange of warm water onto the continental shelf, changes to regional circulation on the shelf, and to the local circulation within ice cavities themselves. Accurately modeling these changes requires high-resolution ocean models in order to link large-scale ocean circulation to sub-shelf melt (Pattyn, 2018; Asay-Davis et al., 2017). Development of these modeling capabilities is still a major focus area of current research. Most coarse-resolution global models do not have ice shelf cavities, and those that do, have large uncertainty regarding changes to the influx of relatively warm Circumpolar Deep Water (CDW) mass into the ice shelf cavities. Ice loss resulting from CDW intrusions has already been observed in the Amundsen Sea region in West Antarctica (Hellmer et al., 2017; Pritchard et al., 2012). One mechanism that has been identified to enhanced CDW import to the Amundsen region is an anthropogenically forced shift in the direction of shelf-break waters (Holland et al., 2019). Mass loss in the Totten region in East Antarctica has also been linked to ocean circulation changes (Greenbaum et al., 2015; Wouters et al., 2015). There is further uncertainty in how ocean eddies — which are unresolved by current standard-resolution climate models — transport heat to the Antarctic coast (Paolo et al., 2015; Stewart and Thompson, 2015). Finally, outside of basal melt uncertainty, there is deep uncertainty in glaciological dynamics (ie. no consensus on what processes to include in an uncertainty analysis, nor how). An example of this is whether ice fracture mechanics, such as the marine ice cliff instability (MICI), could dramatically accelerate ice loss (DeConto and Pollard, 2016; Edwards et al., 2019).

Efforts by the scientific community have surged in hopes of constraining the uncertainty bounds on future SLR from Antarctica (e.g. initMIP-Antarctica (Seroussi et al., 2019) and ISMIP6 (Seroussi et al., 2020)). The typical approach is to run large ensembles of ice sheet model simulations, perturbing different parameters for each run, and then estimating uncertainty based on the model spread (e.g. Golledge et al., 2015; DeConto and Pollard, 2016). Less conventional techniques have recently been

applied, including the use of reduced statistical models (Kopp et al., 2016; Mengel et al., 2016; Fuller et al., 2017; Le Bars et al., 2017) or structured elicitation studies (Kopp et al., 2014; Little et al., 2013; Bamber and Aspinall, 2013). Others have used semi-empirical dynamical models relating GMSL change to global temperature (e.g. Grinsted et al., 2010; Mengel et al., 2016; Kopp et al., 2016). Others still have used very simple reduced-form mechanistic models such as the BRICK (Building blocks for Relevant Ice and Climate Knowledge) model (Wong et al., 2017) to simulate changes in global mean surface temperature and sea level as a function of perturbations to radiative forcing. Despite this assortment of methods, there is still deep uncertainty in how the ice sheet itself will respond to forcing in the future (Bakker et al., 2017).

## 1.2 Benefits of statistical emulation of ice sheet models

Kopp et al. (2017) notes that "Ideally, the integration of process models into probabilistic frameworks...would involve the development and use of fast models – or fast statistical emulators of more complex models – in a mode that allows Monte Carlo sampling of key uncertainties and the conditioning of uncertain parameters on multiple observational lines of evidence. The development of such fast models or model emulators is an involved task." Statistical emulators, sometimes referred to as surrogate models, can be used to fully explore parameter space that would otherwise be too computationally intensive for a process-based model. A typical statistical emulation approach is a response-surface formulation — discussed further in Section 2.3 — such as a Gaussian process or neural network, which interpolates the outputs of a perturbed-parameter ensemble of model runs across its input space (Sacks et al., 1989). Because high-fidelity numerical models are computationally expensive, only a limited number of simulations are typically available for the purposes of uncertainty characterization. The goal of an emulator is to inexpensively predict, from a small training ensemble of an expensive computer model, the output that the model would produce if it were run at a new input setting that was too expensive to simulate. The emulator can then be run in much larger ensembles than the original numerical model in order to explore uncertainties.

Statistical emulators have been used in climate science for some time. For example, Hauser et al. (2012) trained and built a Bayesian artificial neural network with GCM output, using their emulators to calibrate climate models against seasonal climatologies of temperature, pressure and humidity. This generated statistically rigorous probabilistic forecasts for future climate states. Not long after, groups began to apply such statistical methods to ice sheet models as a step toward building ice mass loss projections which included uncertainties. For example, Chang et al. (2014) used spatially-resolved synthetic observations (with data-model fusion) to create a probabilistic calibration of a Greenland Ice Sheet model. Recently, emulation has gained even more traction in the Antarctic ice sheet modeling community. Pollard et al. (2016) used a Bayesian technique involving Gaussian process-based emulations and calibration to provide SLR envelopes based on a 3D hybrid ice-sheet model applied to the last deglaciation of WAIS ($\sim$ 20,000 yr ago). Bulthuis et al. (2019) built an emulator of the continental ice sheet response to a comprehensive set of uncertainties over the next millennium using the Elementary Thermo-mechanical Ice Sheet (f.ETISh) model (Pattyn, 2017) at 20 km resolution. Edwards et al. (2019), built an emulator based on the ice sheet modeling (at 10 km resolution) of DeConto and Pollard (2016) in order to generate probabilistic projections for the Antarctic contribution to sea level rise.

In this paper, we build on these methods by constructing, validating and testing an ice sheet emulator based on the state-of-the-art Community Ice Sheet Model (CISM) version 2.1 (Lipscomb et al., 2019). Our work is novel primarily in its focus on ocean forcing uncertainty in combination with high-resolution glaciological modeling. Specifically:

- We focus on uncertainty in ocean forcing, which is considered the most powerful driver of Antarctic sea level rise in the coming centuries.

- We use realistic transient trajectories of basal melt rates that can be mapped back to ocean models, as opposed to more stylized forcings.

- Statistical emulation is applied to the input uncertainty of the (ocean) climate forcing to the ice sheet model, as opposed to the ice sheet model's parameters.

- We use the Community Ice Sheet Model (CISM) at higher resolution (4km) than previously used to construct ice sheet emulators. Benefits of higher resolution ice sheet modeling include, but are not limited to, improved representation of grounding line locations and complex bedrock topography.

- CISM is spun-up in such a way that it is in steady state with the current climate conditions. As such, any forward runs are divorced from issues of drift, so the response can be attributed to forcing and not to internal ice sheet variability.

There are several advantages to our approach of statistically emulating an ice sheet model's response to ocean forcing uncertainty. Conventional approaches to providing basal melt boundary conditions to a standalone ice sheet model include: (1) parameterizing melt rates from the thermal forcing of a global climate model (Naughten et al., 2018; Golledge et al., 2019; Seroussi et al., 2020; Jourdain et al., 2019), (2) using basal melt rates calculated from regional ocean models with ice shelf cavities (Cornford et al., 2015; Timmermann and Hellmer, 2013), or (3) providing stylized forcings such as instantaneous or linear ramp melt trajectories, such as those found in the MISMIP+ and SeaRISE community experiments (Asay-Davis et al., 2016; Bindschadler et al., 2013). Each choice is tied to a specific set of physical/modeling assumptions, and typically does not probe the deep uncertainties in these assumptions.

Our statistical emulation method, by contrast, is intended to overcome some of these limitations: We design an ice sheet model ensemble that densely samples a wide range of possible basal melt trajectories (a "space-filling" sampling strategy, to be discussed), initially without consideration of where these trajectories come from or which trajectories are most physically plausible. After constructing a statistical emulator of this ensemble, we can then provide the emulator with basal melt assumptions derived from a number of ocean/climate model combinations. This has two main advantages: (1) If we change our assumptions about future basal melt rates — due to expert disagreement, new scientific discoveries, or simple sensitivity analysis — we can interrogate the emulator to obtain new Antarctic discharge and sea level rise projections, without having to construct a new set of ice sheet model simulations with new ocean forcings. A corollary is that we can use the emulator as a component in a modeling pipeline that can predict new sea level rise distributions (and their downstream impacts such as coastal flooding) with respect to uncertainties in large-scale climate forcing. (2) We can sample the emulator as many times as we wish, allowing a

more complete uncertainty characterization of the distribution of future sea level rise, including extensive sampling into the societally relevant low-probability, high-risk tails of the distribution.

## 1.3 Outline

The remainder of this paper is structured as follows. In the Methods section, we describe the ice sheet model configuration, ensemble design and the methods to build and validate our Gaussian process emulator. We present results for the CISM ensemble, and then show a simple example of using the emulator to generate a probability distribution function (PDF) of sea level rise by propagating prior distributions of basal melt rate parameters based on fits to two ocean models under the A1B emissions scenario. Figure 1 shows a schematic of the tasks necessary to design, build, validate, and test the Gaussian process emulator.

## 2 Methods

### 2.1 Ice sheet model configuration

We use CISM, a state-of-the-art 3D, parallel, thermo-mechanical model that runs on a regular mesh grid using a mixture of finite-difference, finite-volume and finite-element methods. CISM has participated in various ice sheet model intercomparisons (e.g. MISMIP+ (Cornford et al., 2020), LARMIP (Levermann et al., 2020), ABUMIP (Pattyn et al., 2019), and ISMIP6 (Nowicki et al., 2020; Seroussi et al., 2020)), and its output was comparable to other higher-order ice sheet models, some that use resolutions of 1 km or higher in the region containing the grounding line. For the experiments described here, the model was run with the following options:

- A depth-integrated higher-order solver based on Goldberg (2011).

- A basal sliding law based on Schoof (2005), in the power-law limit where the effective pressure is equal to the ice overburden pressure.

- Grounding line parameterizations for basal shear stress and basal melt rate (Leguy et al., 2014, 2020).

- Application of basal melting to partially floating cells in proportion to the floating fraction of the cell, which is diagnosed from the thickness and basal topography as part of the grounding-line parameterization.

- A no-advance calving criterion that holds the calving front close to its observed location.

- Surface mass balance (SMB) from late $20^{th}$ century simulations with the RACMO2 regional climate model (van Wessem et al., 2018). SMB is held constant using the RACMO2 1976-2016 climatology in the spin-up and forward runs.

- Geothermal heat flux from Shapiro and Ritzwoller (2004).

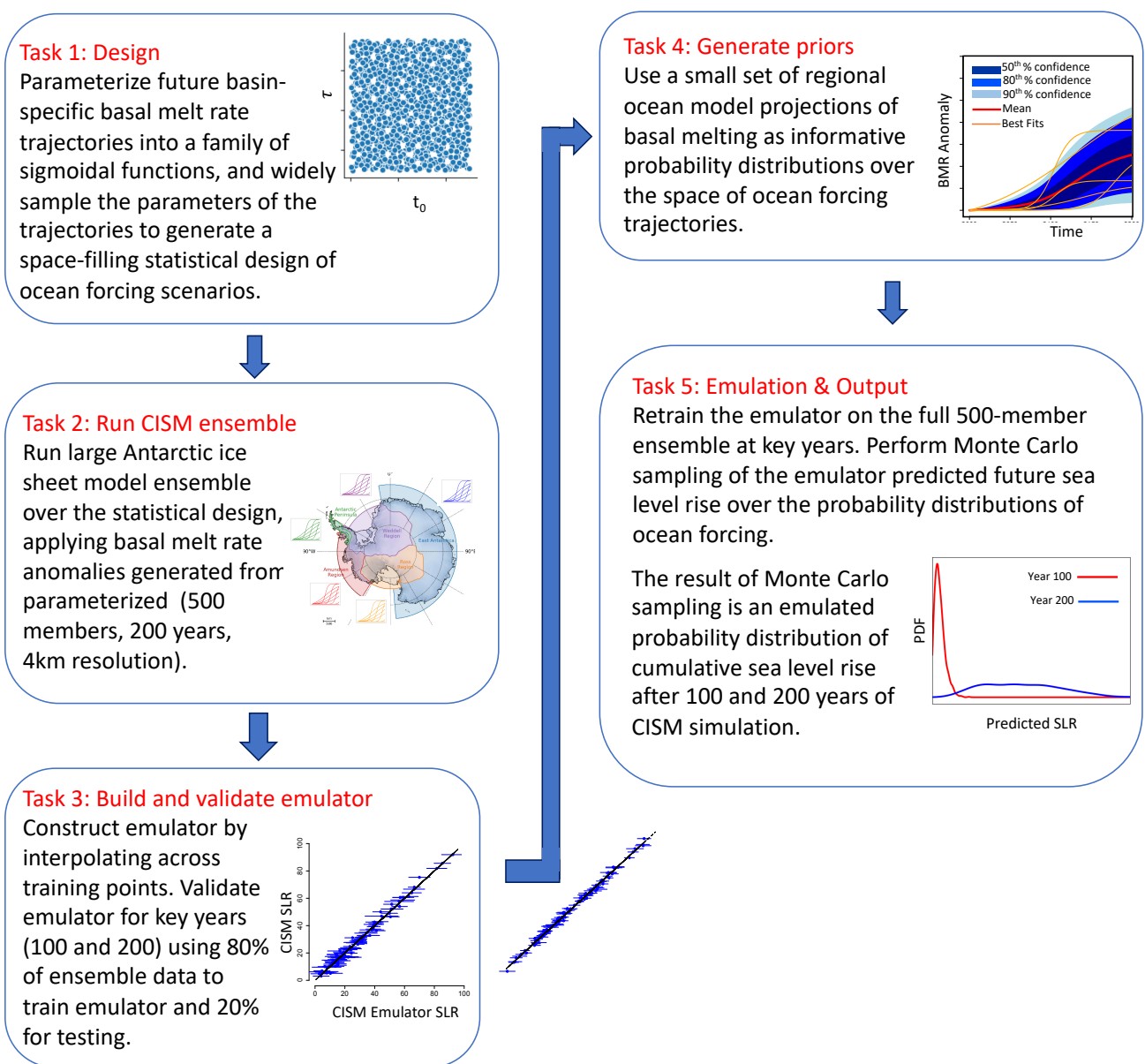

**Task 1: Design**
Parameterize future basin-specific basal melt rate trajectories into a family of sigmoidal functions, and widely sample the parameters of the trajectories to generate a space-filling statistical design of ocean forcing scenarios.

**Task 2: Run CISM ensemble**
Run large Antarctic ice sheet model ensemble over the statistical design, applying basal melt rate anomalies generated from parameterized (500 members, 200 years, 4km resolution).

**Task 3: Build and validate emulator**
Construct emulator by interpolating across training points. Validate emulator for key years (100 and 200) using 80% of ensemble data to train emulator and 20% for testing.

**Task 4: Generate priors**
Use a small set of regional ocean model projections of basal melting as informative probability distributions over the space of ocean forcing trajectories.

**Task 5: Emulation & Output**
Retrain the emulator on the full 500-member ensemble at key years. Perform Monte Carlo sampling of the emulator predicted future sea level rise over the probability distributions of ocean forcing.

The result of Monte Carlo sampling is an emulated probability distribution of cumulative sea level rise after 100 and 200 years of CISM simulation.

**Figure 1.** Schematic showing step-by-step tasks employed in this emulation study – from experimental design to probabilistic sea level rise output. Figure in Task 2 is adapted from Levermann et al. (2020).

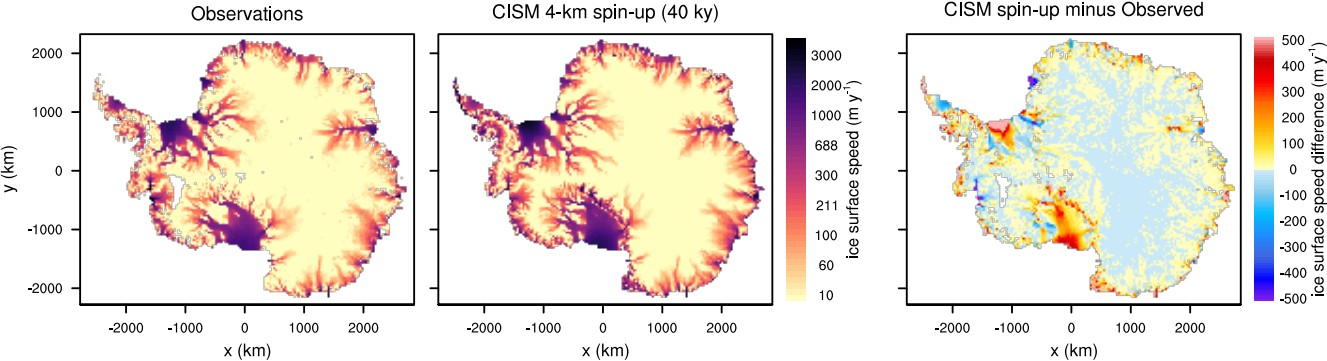

**Figure 2.** Observed (left panel) (Rignot et al., 2011) and modeled (middle panel) Antarctic surface speed (m/yr, log scale) at end of spin-up. Right panel shows the difference between modeled and observed surface speeds (m/yr). White patches represent missing data.

The model is spun up over 40,000 years, with the modeled ice thickness nudged toward observed thickness by adjusting a 2D basal friction parameter field beneath grounded ice and basal melt rates beneath floating ice. This inversion scheme is similar to that of Pollard and DeConto (2012)) and was used for the CISM contribution to initMIP-Antarctica (Seroussi et al., 2019) and the ISMIP6 projections (Nowicki et al., 2020; Seroussi et al., 2020). We note that there is no hydrology in the basal friction field, and the basal melt field is noisy, compensating for other errors in the model or observations. However, this work is a proof-of-concept, and the emulator techniques used here would apply equally well to simulations with more realistic physics. Nudging is strong in the first half of the spin-up, and then tapered off for the second half. The ice thickness gradually approaches a quasi-steady state as basal friction parameters and internal temperatures evolve. The model is run on a uniform 4 km grid, resulting in a spun-up state with good agreement between observed and modeled surface velocity (Fig. 2), ice shelf extent, and ice thickness (Fig. 3), except in regions that are known to be out of steady state, such as the Amundsen sector and the Kamb Ice Stream.

A control run starting from the end of the spin-up and going forward 1000 yr (not shown) shows that there is very little drift (< 1 Gt/yr) in the ice sheet mass. Thus, most changes in ice thickness will be a result of forcing as opposed to internal variability or model drift. This is not fully realistic, since the real ice sheet is never truly in equilibrium with the climate, particularly if current observations are used to tune the model.

Therefore, henceforth, we do not explicitly state the year corresponding to SLR projections. Rather, we refer to our SLR projections as relative to the number of years run forward in the model from the end of spin-up. As a result, the sea level rise projections are not tied to a particular year in the future. Rather, they are meant to show that the emulator is a powerful and useful tool, and SLR predictions are considered a proof-of-concept.

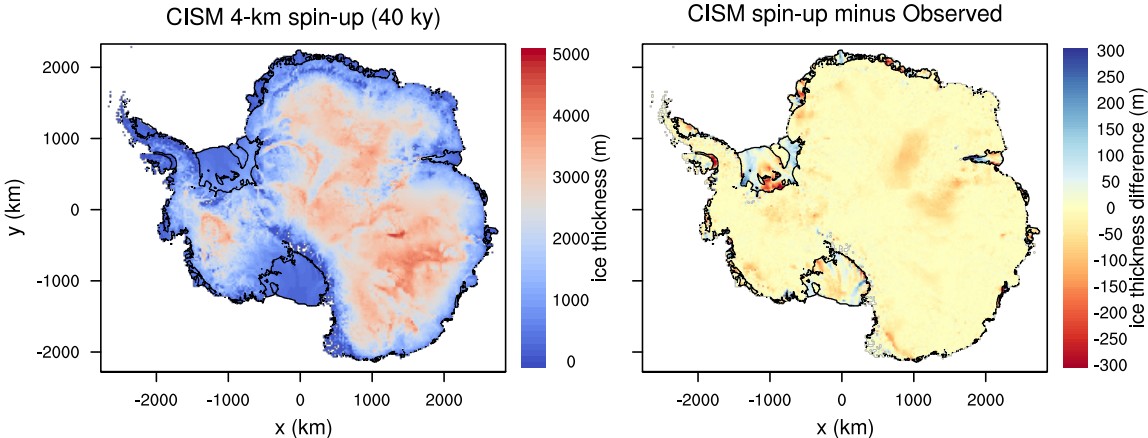

**Figure 3.** Modeled ice thickness (m) (left) and difference between modeled and observed ice thickness (right). Observations are from the BedMachine Antarctica data set (Morlighem et al., 2020).

## 2.2 Ensemble Design

We use ocean model data from Timmermann and Hellmer (2013) and Cornford et al. (2015) (sources described in Table 1) to inform the types of possible basal melt rate trajectory shapes for 200 years forward. The forcing data for the ocean models are generated with the global climate model HadCM3 (Gordon et al., 2000; Collins et al., 2001) under the A1B emissions scenario. A1B is a moderate scenario similar to Representative Concentration Pathway 6 (RCP6). This is then dynamically downscaled by two high-resolution atmosphere models (RACMO2 and LMDZ4) and one ocean model: the high resolution Finite-element Sea ice-ocean model (FESOM) (Wang et al., 2014). We note that the availability of ocean data, particularly melt rate data beyond 2100, was very limited when this study was carried out. Furthermore, while available, output from the Bremerhaven Regional Ice-Ocean Simulation (BRIOS) ((Timmermann et al., 2002)) ocean model was not used in this study because it could not be satisfactorily characterized with parameters in the bounds of our emulator. More detail on this issue can be found in the Discussion.

For each CISM ensemble member, we apply a unique basal melt rate anomaly to each of five basins, following the LARMIP region delineation (Levermann et al., 2020). The regions are the Antarctic Peninsula, Weddell Region, East Antarctica, Ross Region, and Amundsen Region (Fig. 4). We find that we can accurately capture the behavior of all modeled melt rate shapes with a sigmoidal function. Figure 5 shows that the sigmoids (colored curves) do an excellent job of characterizing the variety of shapes seen in the ocean model data in each basin. The equation describing the sigmoid is:

$$M\left(t\right) = \frac{A}{\left(\left(1 + e^{x}\right)\right)} - B \tag{1}$$

where

$$x = -\frac{(t - t_0)}{\tau} \tag{2}$$

$$A = \frac{K}{(K - 1)} \cdot M_{max} \tag{3}$$

$$K = 1 + e^{\left(\frac{t_0}{\tau}\right)} \tag{4}$$

$$B = \frac{A}{K} \tag{5}$$

Equation 1 is a function of three independent parameters: $t_0$ (inflection point of turnover), $\tau$ (timescale of turnover), and
$M_{max}$ (melt rate to which the function asymptotes). However, because we are only able to constrain the melt rate 200 years
into the simulation (we do not have simulations that go out to infinity), we must invert Eq.1 to be in terms of the melt rate at
year 200 of the simulation, $M_{200}$. Using Eqs. 1-5 we can derive coefficients $A$ and $B$ as a function of $M_{200}$. In doing so, we
are able to describe the basal melt trajectories in terms of three parameters: $t_0$, $\tau$ and $M_{200}$. The emulator is built on these three
parameters.

| Region | Ocean Model Source | |
| --- | --- | --- |
| | Timmermann & Hellmer (2013) | Cornford et al. (2015) |
| EAIS | Fimbul (HadCM3/FESOM) | |
| | Amery (HadCM3/FESOM) | |
| Ross | Ross (HadCM3/FESOM) | Ross (HadCM3/FESOM) |
| Amundsen | Abbot (HadCM3/FESOM) | Amundsen Sea Embayment (HadCM3/FESOM) |
| | Pine Island Ice Shelf (HadCM3/FESOM) | |
| | Getz (HadCM3/FESOM) | |
| Peninsula | Larsen (HadCM3/FESOM) | |
| | GeorgeVI (HadCM3/FESOM) | |
| Weddell | Filchner-Ronne Ice Shelf (HadCM3/FESOM) | Filchner-Ronne Ice Shelf(HadCM3/FESOM) |
| | East Weddell Ice Shelf (HadCM3/FESOM) | |

**Table 1.** Sources of ocean model output used in this study. Brackets include specifications on global climate model and regional ocean model.

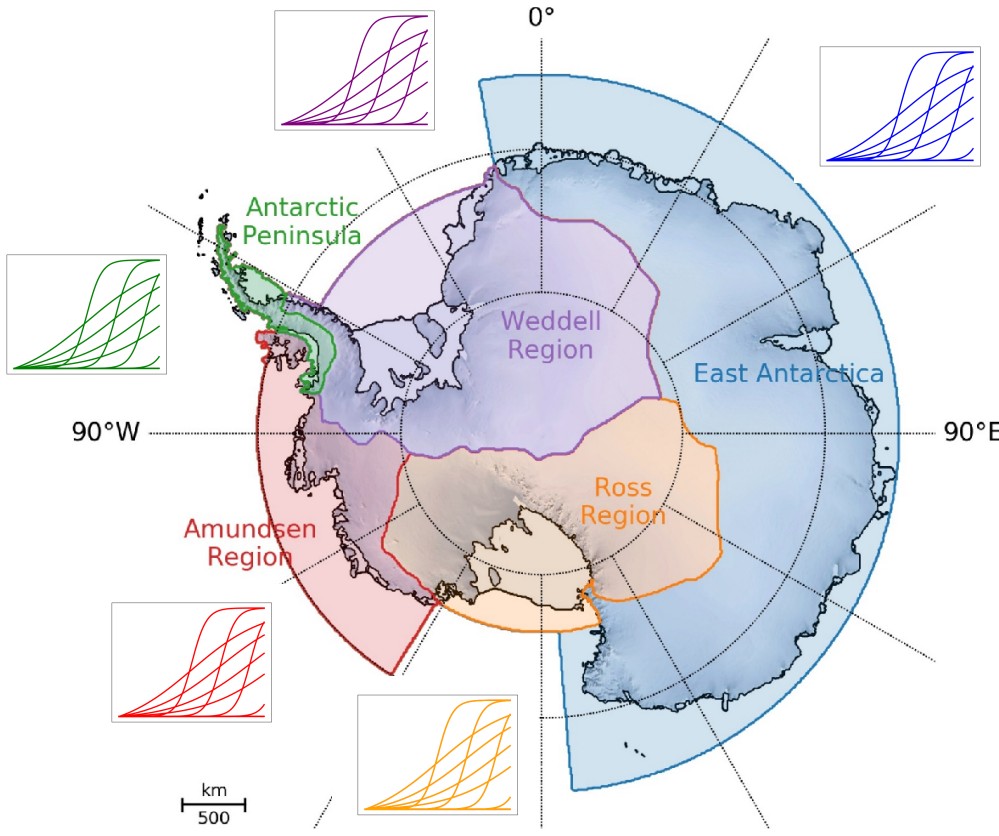

**Figure 4.** LARMIP regions to which basal melt rate anomalies were applied, adapted from Levermann et al. (2020). Each basin pulls random sigmoid shapes for each ensemble member, with the maximum value at the final year (200) scaled to a basin-specific value, illustrated by the schematized color-coded melt rate trajectories.

We use a quasi-random Sobol' sequence as a space-filling design method for each parameter (Sobol', 1967). Compared to pseudo-random Monte Carlo sampling (independent random sampling from a distribution using a deterministic numerical algorithm), space-filling designs reduce the likelihood of clustering leading to uneven sampling. Sobol' sequences also have an advantage compared to some other common space-filling designs, such as Latin hypercubes (McKay et al., 1979), in that they are sequences designed to fully cover the parameter space at each point, recursively filling the space more densely as points

continue to be added to the sequence. This sequential feature of quasi-random sampling allows us to extend the sequence if we desire more ensemble members, while maintaining a space-filling design. A Latin hypercube, by contrast, is not a sequential design and its size must be specified in advance: more points cannot be later added to an existing design without violating the properties of a Latin hypercube. The Sobol' sequence is sampling from bounded uniform distributions on each parameter to generate the emulator training ensemble.

The ranges for $t_0$ and $\tau$ are determined by maximizing the space-filling properties of the parameters whilst capturing all of the sigmoidal characteristics seen in the modeled ocean melt rate projections in Timmermann and Hellmer (2013) and Cornford et al. (2015). Expert judgement was used to limit the ranges for these to remain 'physically reasonable'. Therefore, using a Sobol' sequence, we sample uniformly between the following lower and upper bounds for the sigmoid-defining parameters: $t_0 \in [100:225]$, $\tau \in [10:75]$, $M_{200} \in [0:1]$. $M_{200}$ is later scaled on a basin-by-basin basis (Figure 4, Figure 5), informed

by literature values of melt rates at the year 2200 from ocean melt rate projections in Timmermann and Hellmer (2013) and Cornford et al. (2015). Specifically, we allow the $M_{200}$ upper bound to be at least twice the maximum value found in the literature for each basin (shown also in Fig. 5). The maximum values imposed for year 200 by basin are therefore:

-   Antarctic Peninsula: 12 m/yr

-   Amundsen Region: 50 m/yr

-   Ross Region: 20 m/yr

-   East Antarctica: 36 m/yr

-   Weddell Region: 16 m/yr

    An example of some basin-specific Sobol' generated sigmoids that would be fed to CISM are shown in Figure 6. For each CISM run, a random basal melt rate curve generated with the Sobol' sequence is chosen for each basin. We have chosen not

to assume any correlation between basins. Of course, general ocean warming occurs with global warming, but there may be unique regional circulation patterns that could cause very different basal melt rates in one region compared to another.

    We run an ocean-forcing-perturbed CISM ensemble (with 500 members) over the entire Antarctic domain, where CISM surface mass balance is a climatology, and unique ocean melt rate anomalies are added to the background basal melt rates from the end of the CISM spin-up. The melt anomaly is applied to any newly ungrounded cells that appear through the simulation.

Since the spun-up basal melt rates are resolved along the ice draft, the original state of basal melt rates has a realistic character, in that it varies along a 2-D surface that is a function of depth so has spatial variability with increasing basal melt rates near the

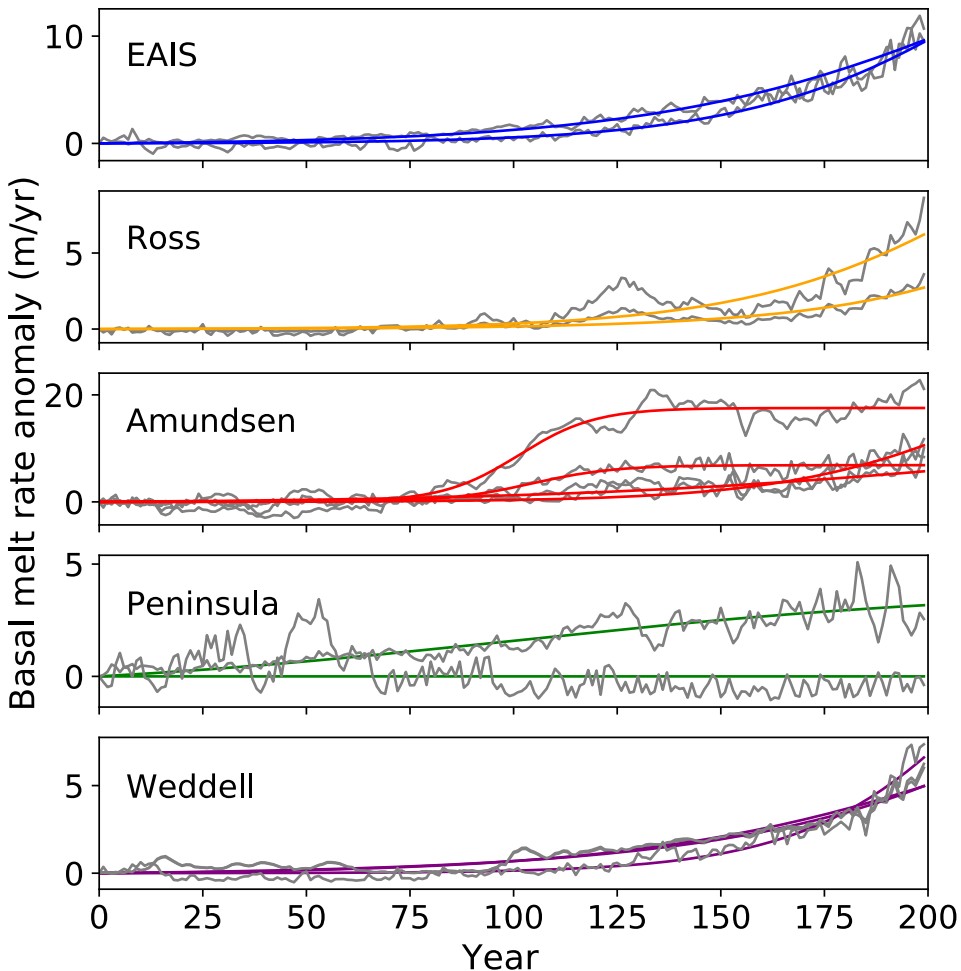

**Figure 5.** Basal melt rate anomaly data from Timmermann and Hellmer (2013) and Cornford et al. (2015) (grey curves) overlaid with best sigmoidal fits (colored curves). Colors correspond to the LarMIP regions in Figure 4. Note the different y-axis range for each region.

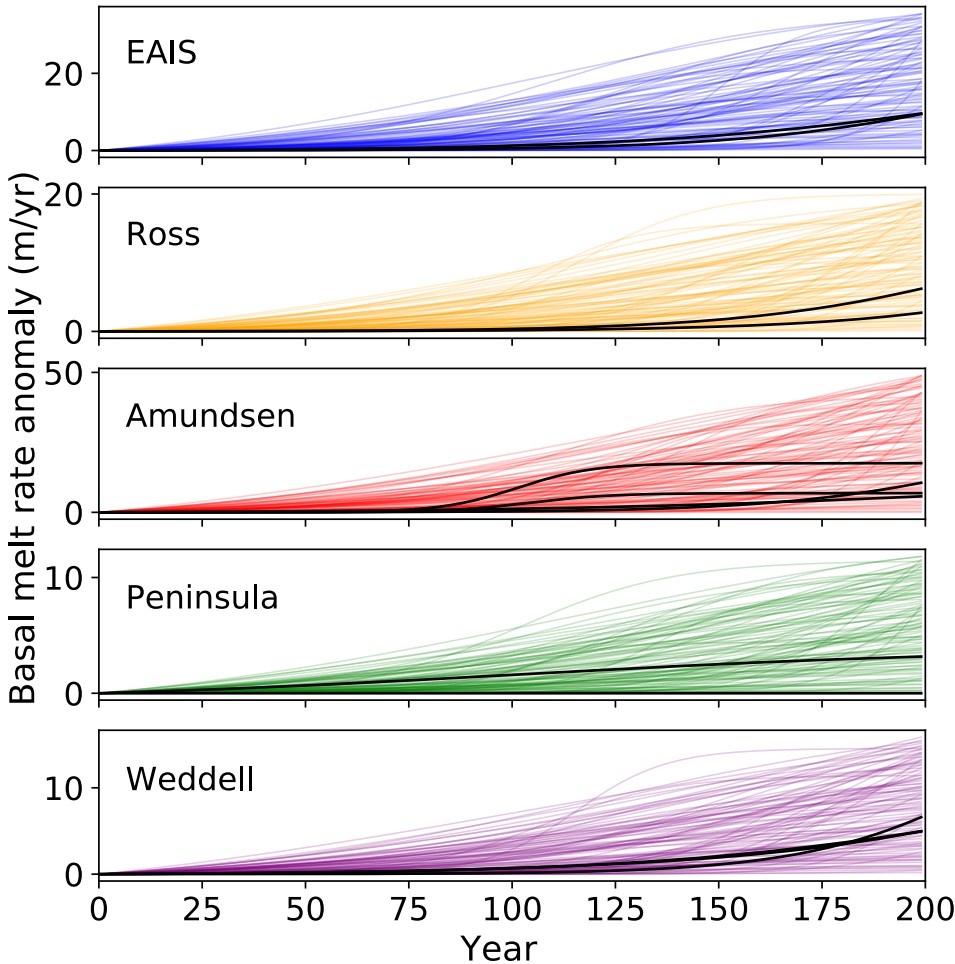

**Figure 6.** Sample of 100 Sobol'-generated melt rate trajectories (colors) and best fits (by least squares optimization) to ocean model data from Timmermann and Hellmer (2013) and Cornford et al. (2015) overlaid in black for each basin. By definition, the Sobol'-generated curves are permitted to sample up to the maximal melt values given in the text, representing roughly twice the maximum 'data' melt rate at year 200.

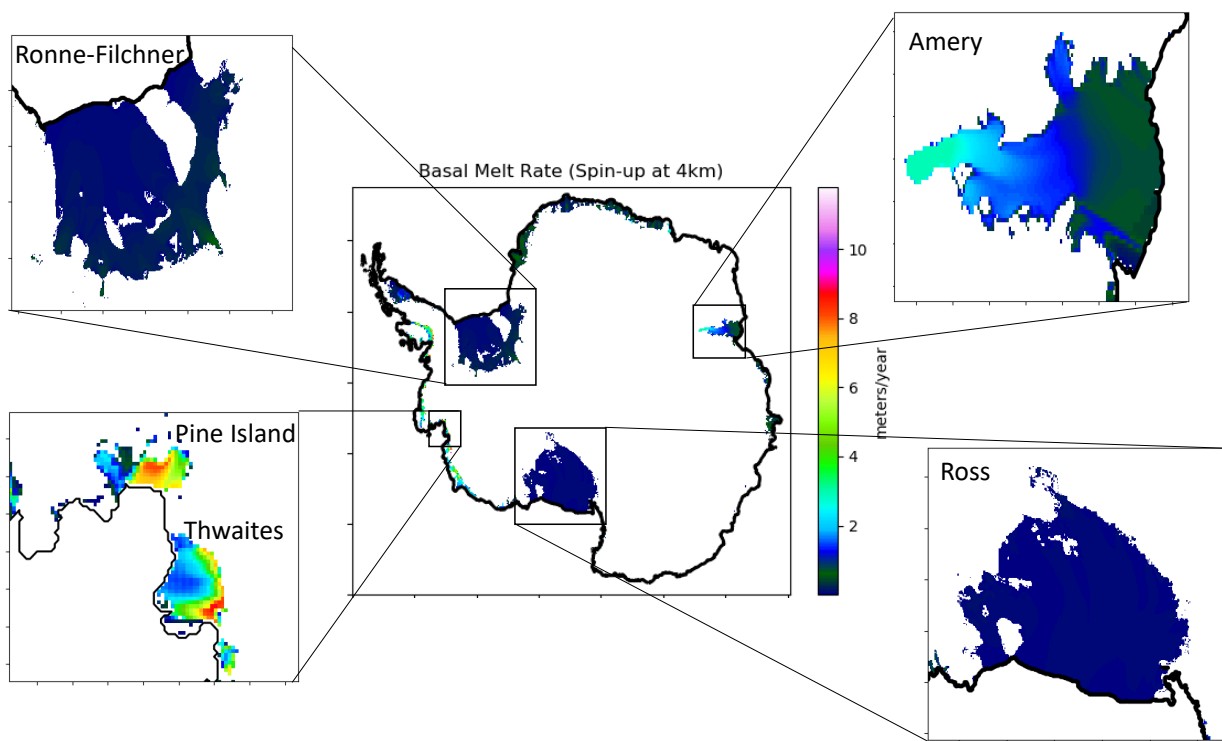

**Figure 7.** Basal melt rate at end of spin-up at 4km resolution for the all floating ice areas. Zoomed regions include the PIG/Thwaites region, Ronne-Filchner, Amery and Ross ice shelves.

grounding line (Fig. 7). We note that by imposing a spatially uniform basal melt rate anomaly for each basin, we are neglecting to account for complex patterns in sub-shelf ocean circulation changes or depth-dependence in the anomaly itself. However, most available melt rate projection data, particularly any estimates that go out 200 years, are regional averages. Furthermore, we must limit the number of parameters we vary in order to be able to run a large enough ensemble to appropriately sample the parameter space and subsequently build an emulator.

## 2.3 Gaussian Process Emulator of the CISM ice sheet model

The emulator constructed here predicts a single output (Antarctic SLR anomaly in a specified year of simulation), as a function of a 15-dimensional input vector representing the perturbed regional basal melt trajectories (5 regions x 3 sigmoid parameters, $(t_0, \tau, M_{200})$). The training set is the full 500-member CISM ensemble. On time scales of a few centuries, the effects of smoothing over the stochastic variability in the forcings with a sigmoidal curve is expected to be minimal (Hoffman et al.,

2019). A Gaussian noise term is added to the emulator prediction to account for natural SLR variability; the standard deviation of natural variability is estimated from the Rignot et al. (2019) Antarctic mass loss dataset ($\approx 1.5$ mm).

Statistical emulation in this paper is of the response surface type (Box and Wilson, 1951). Our training ensemble — consisting of pairs of 15-dimensional input vectors and their corresponding scalar model outputs — can be thought of as samples from a function that maps model inputs to outputs (a response surface). To predict CISM SLR output at a new point in input space, not contained within the training ensemble, a smooth response surface is constructed by interpolating the points in the training set. The emulator prediction for a particular point in input space is the model output interpolated to that point, lying on the response surface.

The statistical emulator used here is of the popular Gaussian process regression family (Sacks et al., 1989), the implementation found in the 'GPfit' R library (MacDonald et al., 2015; Ranjan et al., 2011). We use the standard squared-exponential covariance with independent (factorized) correlation functions for each parameter, and a small nugget for numerical conditioning. The Gaussian process variance hyperparameter is estimated analytically, as is the nugget (following the lower bound given in (Ranjan et al., 2011)), whereas the (reparameterized) correlation length scale parameters are fit by minimizing the negative profile log-likelihood. Generally, Gaussian processes perform nonlinear, multivariate, smooth interpolation of a (potentially irregular) set of training data, which is computed via a statistical regression procedure. Usually the smoothness of data being fit is estimated as part of the interpolating procedure. A Gaussian process's interpolating surface is 'optimal' in a technical sense, going back to the geostatistics literature where it is known as 'kriging' (Matheron, 1962; Krige, 1951). The modern Bayesian interpretation of Gaussian processes provides a second useful statistical feature, namely they can provide the uncertainty in their own predictions, i.e., a built-in estimate of interpolation error. When making SLR predictions, the emulator's interpolation error ('code uncertainty') is added as a second, independent noise term to the Gaussian process emulator's mean prediction. With the inclusion of this error term as well as the natural variability noise terms, the emulator's SLR predictions become a stochastic function of its basal melt inputs, even though the emulator is approximating the output of a deterministic model (CISM).

### 2.3.1 Emulator Validation

To validate the emulator, we randomly withhold 20% of the ensemble members, and build an emulator based on the remaining 80% of SLR data from the CISM ensemble. (For the purposes of SLR projection, we train the emulator on the full ensemble.) The predicted SLR values at years 100 and 200 show good emulator performance with correlation coefficients of 0.98 and 0.99, respectively, against the withheld CISM output (Fig. 8). For year 100, 8% of hold-out validation points lie outside the 2-sigma predictive intervals, which is plausible (one would expect 5%). Based on these highly correlated results, we are confident in the emulator's ability to approximate CISM's SLR output 100 and 200 years into the simulation.

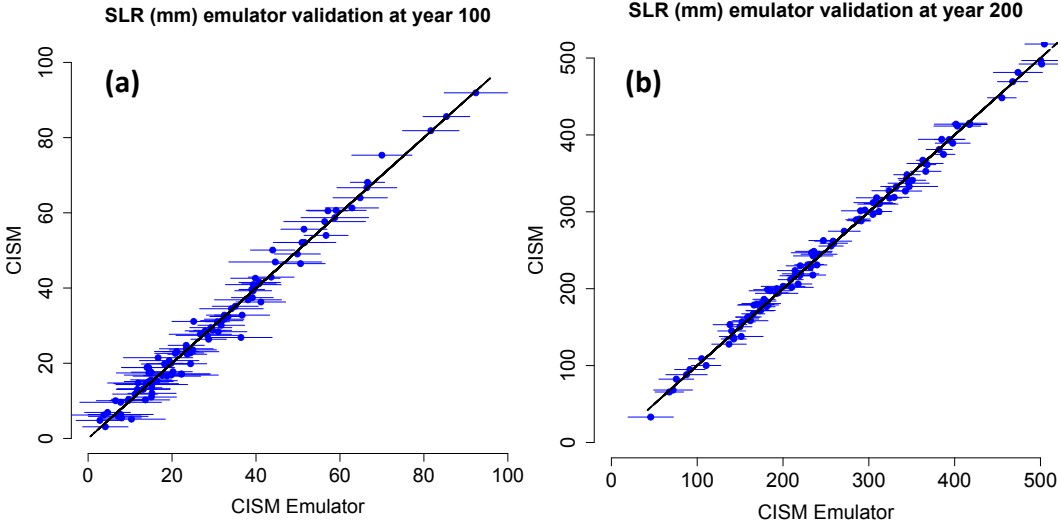

**Figure 8.** Emulator validation: True (CISM) vs. Predicted (CISM Emulator) SLR values for year 100 (a) and 200 (b). Black solid line shows 1:1 correlation. Correlation coefficients are 0.98 and 0.99 respectively. Note the different x and y scales for (a) and (b). Error bars go from $-2\sigma$ to $+2\sigma$.

## 3 Results

In this section we present results from the CISM ensemble, showing the SLR contributions at year 100 and 200. We note that the SLR results from the ensemble itself should not be considered physically realistic as the prior melt parameters are uniformly sampled (as described in the Methods Section).

We also present results from a simple example illustrating the propagation of parametric uncertainties ($M_{200}$, $t_0$ and $\tau$) describing melt rate trajectories (by basin) derived from ocean model projections. The priors are generated with three different, but equally valid, methodologies, and then propagated through the emulator, resulting in a probabilistic SLR prediction for each method at years 100 and 200.

### 3.1 CISM ensemble results

The CISM ensemble consists of 500 members where each member is forced by five melt rate trajectories, one in each basin. Figure 9 shows the sea level rise time series resulting from the full 500-member ice sheet ensemble (blue shading), with the ensemble mean shown in red. The distributions of SLR at year 100 are more constrained (ranging from $0.5 - 96$ mm) than those in year 200 (ranging from $33 - 543$ mm) (Figure 9 inset). The PDFs for the ensemble are shown in Fig. 11 (grey curves). Again, we note that these SLR projections are not physically meaningful since the parameter sampling over which the ensemble is created is uniform. The ensemble is designed purely to be used for the creation of an emulator.

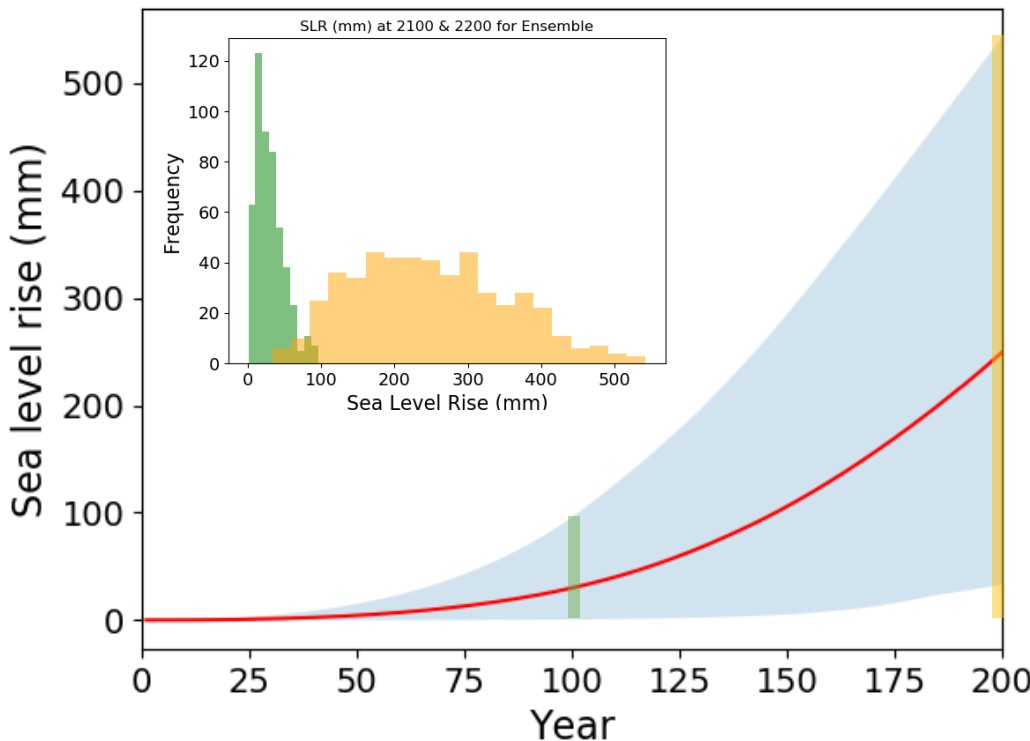

**Figure 9.** 500-member ensemble SLR contributions (mm) from year 0 to 200, distribution (blue shading) and ensemble mean (red curve). Inset shows sea level rise histogram (mm) at year 100 (green) and 200 (orange) of simulations, corresponding to the vertical green and orange lines in top panel. Note the tighter distributions at year 100 than year 200.

## 3.2 Probabilistic Prediction of Antarctic Sea Level Rise

We test the emulator by running three simple examples of prior distributions (based on the FESOM ocean model data described in Timmermann and Hellmer (2013) and Cornford et al. (2015) (sources in Table 1) through the emulator. We produce a probability density function (PDF) of SLR at years 100 and 200 for each of these three methods:

– Method 1: Individual fits + normal distribution

This method is performed by generating a distribution of prior parameters based on the 'best fits'. The best fits are found by a least-squares optimization between the fitted sigmoid curves and the original basal melting rate anomalies from the ocean models (Fig. 5 colored and grey curves respectively). The sigmoid parameters that describe the data fits are then used to generate a normal distribution that serves as the prior. This final step is to allow for the possibility that other ocean models, not considered here, could lead to plausible parameter values. The emulator then samples parameters from

this truncated normal distribution. These prior distributions can be combined and presented as a distribution of sigmoid anomalies (Fig. 10).

- – Method 2: Window fits + direct sampling
This method constructs a windowed set of good parameter values for each ocean model. The window size is defined as 2 SD around the best fit sigmoid. Instead of finding a singular best fit to the ocean model as in Method 1 over which a normal distribution is generated, only fits within this window are used. The windowing is to allow for relaxation away from the edge-hitting parameters (further details on this phenomenon in the Discussion). For each region there is an equal-probability mixture of 'windowed fits' across the ocean models to represent the multi-model uncertainty. This
method does not account for the possibility of melt trajectories not represented by the ocean models.

- – Method 3a: Mixture method (Window fits +normal distribution)
     This method uses a mixture of Methods 1 and 2, an attempt to get the 'best of both worlds': account for non-identifiability/ambiguity in model fits by including a windowed set of good fits as in Method 2, but fit a continuous distribution of the model fits so that probability does not concentrate only on the parameter space locations of the ocean models. This gives nonzero
probability to ocean melt trajectories that don't come from the ocean models in order to account for multi-model uncertainty. So, the same windowing technique is used as in Method 2, but instead of using the parameters of the windowed curves directly as our priors, we generate a normal distribution around the windowed fits as in Method 1. This may be thought of as an approximation to the hierarchical Bayesian approach taken in Jonko et al. (2018), where the parameters arising from fitting each climate model are assumed to be a sample from an underlying multi-model distribution.

– Method 3b: Mixture method (Window fits + multivariate normal distribution)
     This method is the same as Method 3a in that we want to allow for the possibility of other ocean models not contained here. However, unlike in method 3a, it does not assume an independent normal distribution for each parameter. Instead, in order to account for correlation across parameters, we use a multivariate normal distribution (aka a tilted normal).

By propagating the priors generated for each method through the emulator, we can predict SLR probability distributions for
315 year 100 and 200 (Fig. 11) corresponding to each method. An example of how the prior parameters are combined to form a distribution of basal melt rate anomalies is shown in Fig. 10. In general, the priors generated encompass the 'best fits' quite well. The likeliest SLR at year 100 is found to be $\sim 4 - 6$ mm depending on the prior method used. The likeliest SLR in year 200 is $\sim 71 - 82$ mm depending on the prior method used. The mode (likeliest SLR) of the PDF for each prior method, along with the $2.5\%$ and $97.5\%$ SLR values are shown in Table 2.
As explained in the Model Configuration section, we do not consider this to be a prediction for the year 2100 or 2200, simply because the assumption that the ice sheet is in equilibrium with the climate is not fully realistic. In the future, one option to eliminate this problem might be to run a historical simulation before the perturbation simulations in order to bring the model into a more sensitive state. However, preliminary attempts to do this with CISM have been focused on global climate model output, which did not capture the recent melting in the Amundsun Sea. Another reason for the small response is that the melt

rate perturbation is applied uniformly over the basin. Focusing melt near the grounding line (for a given basin mean) would give greater retreat (Lipscomb et al., 2021). Section 4 elaborates on both of these points.

| Prior Method | Year 100 SLR | | | Year 200 SLR | | |
|---|---|---|---|---|---|---|
| | 2.5% | 97.5% | mode | 2.5% | 97.5% | mode |
| Method 1 (Individual fits + normal distribution) | 0.5 mm | 18.7 mm | 6.3 mm | 40.3 mm | 170.8 mm | 82 mm |
| Method 2 (Window fits + direct sampling ) | 0.3 mm | 16.6 mm | 3.9 mm | 35.3 mm | 218.4 mm | 71 mm |
| Method 3a (Window fits + normal distribution) | 0.6 mm | 19.9 mm | 5.3 mm | 41.6 mm | 183.4 mm | 81 mm |
| Method 3b (Window fits + multivariate normal distribution) | 0.4 mm | 14.1 mm | 3.8 mm | 39.7 mm | 159 mm | 75 mm |

**Table 2.** Emulated sea level rise prediction statistics for three prior methodologies.

## 4    Discussion and Conclusions

The goal of this work is an in-depth exploration of statistical methods designed to project the effects of a plausible range of sub-shelf ocean forcing conditions upon Antarctic sea level rise uncertainty. We have presented an emulator-based approach to
330 derive probabilistic projections of Antarctic sea level rise from a large perturbed basal melt rate ensemble of ice sheet model simulations. This work comes on the heels of other community efforts to quantify uncertainties in Antarctic sea level rise. For example, the LARMIP-2 project (Levermann et al., 2020) applies a linear response theory approach to 16 different ice sheet models (including CISM) in order to estimate the uncertainty of Antarctica's future contribution to global sea level rise that arises from uncertainties in ocean forcing. Their method, similar to that in Castruccio et al. (2014), relies on the assumption of
335 linearity in the ice sheet response, which is generally valid for moderate basal melt rates but tends to break down (including in the CISM model) at higher melt rates, particularly after the first century of simulation. Our emulator method, on the other hand, does not rely on a linearity assumption and is thus valid over a very wide range of ocean scenarios, including the stronger forcing regimes. It is in the high-end (tail-area) ocean forcing scenarios where the greatest societal risk lies, so our focus is to carefully represent those accurately. In the future we could consider a more direct comparison of our results to the linear
response approaches used by Levermann et al. (2020).

We designed and ran a 500-member CISM ensemble, perturbing basal melt rates for 200 years over a wide range of possible future melt trajectories, concentrated on trajectories derived from the high-resolution FESOM ocean model under the A1B warming scenario. With this ice sheet ensemble, we constructed and validated a CISM emulator that provides ice retreat as a function of basal melt rate anomalies applied at the coastal shelves. The main advantage of the emulator is that we can use
it to densely sample a wide range of possible basal melt forcing, including the high-risk tails of the basal melt projections. The emulator can produce probabilistic sea level rise projections over much larger Monte Carlo ensembles than are possible by direct numerical simulations alone, thereby providing better statistical characterization of uncertainties. Furthermore, the

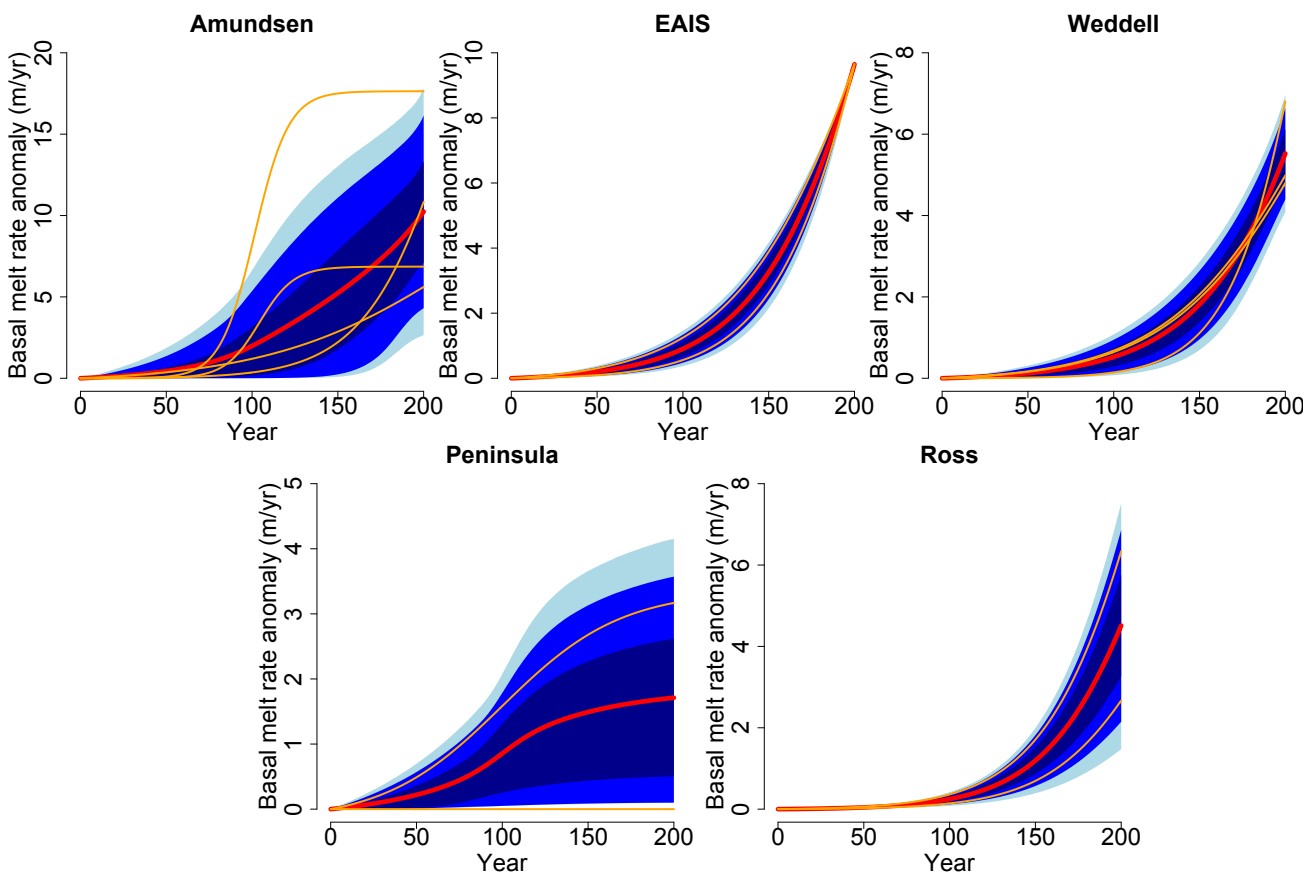

**Figure 10.** Prior basal melt rate anomaly (m/yr) trajectories generated with Method 1 (Individual fits + normal distribution) as seen by CISM for each region: Amundsen Region, East Antarctic ice sheet (EAIS), Weddell Region, Antarctic Peninsula, and Ross Region. Red lines show mean of distribution, blue shaded zones correspond to 50th, 80th and 90th % confidence intervals (from dark blue to light blue respectively), orange lines correspond to best fits to ocean model data. Note the different y-axis range for each panel.

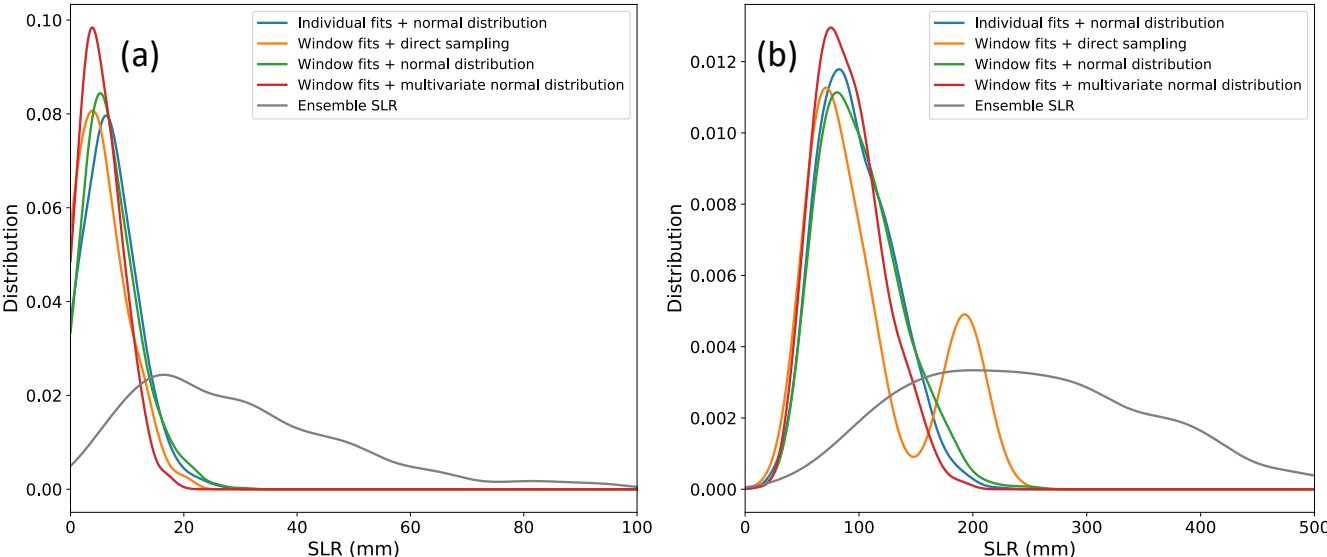

**Figure 11.** Sea level rise probability distributions for (a) year 100 and (b) year 200. The ensemble SLR PDF at year 100 and 200 (grey), and the predicted SLR PDFs for three prior methodologies described: (1) Individual fits + normal distribution (blue), (2) window fits + direct sampling (orange), (3a) window fits + normal distribution (green) and (3b) window fits + multivariate normal distribution (red) .

emulator can be used to predict the simulated ice sheet response (along with associated uncertainty bounds) under different assumptions about basal melt rate probability distributions without running any more dynamic ice sheet simulations. This is especially advantageous as new and updated information becomes available. Although 200-year CISM simulations are relatively affordable at 4-km resolution, this might not be the case at higher resolution; each doubling of resolution leads to about a $10\times$ increase in cost. We have further shown how we can propagate uncertainty through the emulator using different methods of generating ocean melt rate priors under the A1B scenario.

We used multiple, equally valid, methods for sampling priors. The first method (individual fits + normal distribution) finds the best fit to the ocean model output with a sigmoid (within our emulator parameter bounds) and generates a normal distribution around these parameter fits from which inputs are sampled. The most obvious limitation here is the sparse number of priors (ocean model data) available. This results in deriving a continuous probability distribution of melt rate parameters from a very small collection of ocean models. Furthermore, selecting a single 'best' fit found by least-squares optimization leads, in some cases, to parameter estimates on the boundary of the plausible range of values. For the most part, we do not believe these edge-hitting fits imply that the parameter bounds must be expanded. Even if we expand the parameter ranges, the optimizer still moves along flat ridges of the loss function and hits the boundaries of whatever new ranges we impose (not shown). These edge-hitting fits are largely an artifact of non-identifiability between the sigmoid parameters, rather than misspecification / discrepancy of the sigmoid model of basal melt rate trajectories, or too-narrow bounded priors.

In order to account for non-identifiability, we included two other methods that did not just use one best fit, but rather allowed for many fits in a window of width 2SD around the best fit sigmoid. This is to allow for relaxation away from the edge-hitting parameters. The second prior method (window fits + direct sampling) calculates hundreds of fits that fall within each window. These then serve as the sampled inputs to the emulator, without any assumption of distribution beyond them. This windowing method increases the number of parameter prior samples per ocean model and includes fits that do not hit parameter boundary edges. The direct sampling limits the prior sampling strategy to 'more likely' spaces as opposed to assuming a distribution of likelihood between ocean model realizations. Methods 3a (window fits + normal distribution) and 3b (window fits + multivariate normal distribution) are a mixture of methods 1 and 2, and meant to represent the 'best of both worlds'. The windowing technique is used to generate hundreds of samples per ocean model, over which a normal distribution is generated. The reason we take the last step, instead of using the mixture-of-windows directly, is to allow for the possibility that other ocean models, not considered here, could lead to plausible parameter values not contained within the windows for any of the ocean models. By smoothing over the mixture of windows, we assign a nonzero probability to settings that lie near, but not within, the window from any given ocean model. This may be thought of as an approximation to the hierarchical Bayesian approach taken in Jonko et al. (2018), where the parameters arising from fitting each climate model are assumed to be a sample from an underlying multi-model distribution. We include a multivariate normal distribution (Method 3b) in order to account for correlation across parameters. This is our preferred method as it is the most principled approach.

Over a range of future melt-rate trajectories derived from a small collection of high-resolution regional ocean models, the emulated CISM model projects from 0.3 mm $(2.5\%)$ to 20 mm $(97.5\%)$ of Antarctic SLR in year 100 and 35 mm $(2.5\%)$ to 218 mm $(97.5\%)$ in year 200. The likeliest SLR at year 100 is found to be $\sim 4-6$ mm (depending on prior method), which falls within the range of century-scale future projections of sea level rise from Antarctica in the literature, albeit on the low end of most estimates. The likeliest SLR in year 200 is $\sim 71-82$ mm. The likeliest predicted SLR in both of these years is therefore not strongly dependent on the prior method choice. Prior methods 2 (window fits + direct sampling) and 3b (window fits + multivariate normal distribution) produce the lowest SLR prediction for year 100. As expected, by using a multivariate normal (method 3b) instead of a normal distribution (method 3a), the SLR prediction shifts closer to the direct sampling (method 2) prediction which also implicitly has correlations in it. A notable difference in method 2 and method 3b, however, is that method 2 results in bimodality in year 200 (Fig. 11). This is an artifact of sampling over a small discrete set of ocean models with no sampling of the parameter space between models. There is a bimodality for the same reasons in the year 100 prediction for this method as well, but it is smoothed out when emulator uncertainty is accounted for. We caution that these SLR results should be interpreted as a proof-of-concept of a method to quantify SLR uncertainty with respect to uncertainties in ocean forcing, rather than a reliable SLR projection tool at this point. There are several avenues for improvement.

The most obvious place for improvement is to increase the ensemble size to expand the parameter range. While non-identifiability is the likely issue in most edge-hitting sigmoid fits, we have evidence that for one ocean model (BRIOS) there were no parameters within our prior range that could generate good fits. The physical origin of the misfit is that our range did not allow for earlier $t_0$ values that correspond to earlier inflection points in the curve. Therefore, this model was excluded from our simple prior propagation examples. The optimal solution would be to run more ensemble members in order to expand our

parameter range, but computational time was no longer available. We identify this as a major limitation of this work, however we note that the solution would be straightforward given the resources.

Another important limitation of this study is that we apply a uniform melt rate perturbation to an entire basin, neglecting to account for melt rate depth-dependence. This assumption could be relaxed in future work, for example, if we were to add (or swap a sigmoid parameter with) another parameter for ice-ocean physics parameters that control the extent to which melting is focused near the grounding line. Further improvements would be to increase the ice sheet model resolution, use more realistic melt parameterizations, and include novel physical mechanics such as hydrofracture and cliff collapse (DeConto and Pollard, 2016). Eventually, the ice sheet emulator could be included in a larger system linking different sources of uncertainty with multiple emulators, such as a SLR/coastal flooding integrated assessment. As noted previously, the spin-up we use is advantageous because it is in equilibrium with the modern forcing and therefore allows us to isolate the effects of forcing as opposed to model drift in forward runs. However, the drawback of such a spin-up is that one expects the present-day ice sheet to be out of equilibrium with present-day ocean forcing. Therefore, the model could have a lag in the response to basal melt rate anomalies, and therefore underestimate future sea level rise.

One way to address this shortfall in future work might be to insert a 'historical forcing' to the spin-up in order to ramp up the ice sheet to a more sensitive state before applying further basal melt anomalies. Of course, there is still the open question of how best to parameterize basal melt rates. Incorporating some poorly constrained parameters from current melt rate parameterizations (e.g. Favier et al., 2019; Jourdain et al., 2019) into the parameter space could be one way to begin unraveling the uncertainties associated with the parameterizations themselves. Further, the issue remains of how open ocean waters enter and mix under ice shelves. Despite increased observations and measurements of the Southern Ocean properties in recent years (Newman et al., 2019), sub-shelf circulation and melt processes are still largely unknown in the existing glaciological literature that focuses on land ice dynamics (Ritz et al., 2015; Pollard et al., 2016).

Another issue we encountered was the lack of data available to use as future basal melt rate tajectories. This was particularly difficult given our attempt at projecting out 200 years instead of the standard 100 years where most CMIP5/CMIP6 simulations terminate. One possibility to collect more priors is in the case of moving to a thermal forcing spin-up, such as in Lipscomb et al. (2021). In such a scenario, the ocean forcing consists of thermal forcing anomalies (derived from ocean temperature and salinity) instead of basal melt rates. The ice sheet model has a parameterization to convert thermal forcing to melt rates instead of being given melt rates directly. This approach was used for ISMIP6 Antarctic projections. With such a method, one could use much more data from the CMIP6 dataset, particularly since the ISMIP6 effort has already generated thermal forcing files for certain projections/models (Jourdain et al., 2019). Again, these data cease at year 2100 but would be a convenient starting point.

The general issues described above, particularly how to move from coarse (global climate model scale) knowledge of ocean temperatures to higher resolution sub-shelf melt rates, is a confounding issue in the current state of Antarctic ice sheet modeling. We need global *and* regional ocean models to help address how ocean circulation will change, as well as how eddies transport water from the open ocean to the continental shelf. Not only do we not know how general increases in ocean temperatures will translate to sub-shelf melt rates, but also changes in ocean circulation could impact the transport of relatively warm water to

the continental shelf, thereby increasing sub-shelf melt rates as well (e.g. CDW). Progress in this direction will require larger

ensembles of high-resolution regional and global ocean models that sample a wide range of climate scenarios driving Southern Ocean circulation change and variability. Indeed, regional ocean models would be a good target for emulation since these models are particularly expensive, especially when run beyond a few decades.

*Code and data availability.* Code and Data are available at: https://github.com/mberdahl-uw/EmulatorPaper

## Appendix A

440 **A1**

*Author contributions.* MB and NU designed the experiments, with input and advice from GL and BL. GL and MB staged the experiments and MB ran them. BL developed and provided an ice sheet spin-up. NU wrote the statistical model that was run by MB. MB prepared the manuscript with contributions from all co-authors.

*Competing interests.* The authors declare that they have no conflict of interest.

*Acknowledgements.* Mira Berdahl was supported by the U.S. Department of Energy (DOE) Office of Science (Biological and Environmental Research), Early Career Research program. This material is based upon work supported by the National Center for Atmospheric Research, which is a major facility sponsored by the National Science Foundation under Cooperative Agreement No. 1852977. This research used resources provided by the Los Alamos National Laboratory Institutional Computing Program, which is supported by the U.S. Department of Energy National Nuclear Security Administration under Contract No. 89233218CNA000001. Further computing and data storage resources, 450 including the Cheyenne supercomputer (doi:10.5065/D6RX99HX), were provided by the Computational and Information Systems Laboratory (CISL) at NCAR. We thank Matthew Hecht for keeping continuity in this work while MB was on maternity leave. MB would like to thank Eric Steig for mentorship and logistical support during this work.

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
