# Peer review of "Statistical emulation of a perturbed basal melt ensemble of an ice sheet model to better quantify Antarctic sea level rise uncertainties"

_The Cryosphere, 2020_

## Referee Comment (RC1) · Anonymous Referee #1 · 4 Sep 2020

**Reviewer response to the manuscript: 'Statistical emulation of a perturbed basal melt ensemble of an ice sheet model to better quantify Antarctic sea level rise uncertainties'**

Anonymous

September 4, 2020

Based on a large ensemble of the state-of-the-art Community Ice Sheet Model simulations the authors develop a flexible and efficient method to represent the Antarctic ice sheet under a wide range of ocean melt forcings, a major part of the general climate forcing in Antarctica. The manuscript explores the use of an statistical emulator for updated projections of 200-year sea level rise contributions. The link between global climate predictions and the Antarctic response is a key challenge the ice sheet modelling community is currently facing and the use of advanced uncertainty quantification methods, as done here, is likely to play an important role for this.

The presented work is therefore of very high relevance, however, I have concerns regarding the proposed approach to find probabilistic projections and a lack of testing of this approach. These are largely evolving around refining the parameter priors which should make it easier to address my concerns for revision. Once this is done I see the potential for follow up studies to combine multi-GCM southern ocean climate prediction distributions with ocean melt prediction PDFs and the ice sheet model sea level emulator developed here. This could be an major step towards a quantification of the sea level rise projection uncertainties. I will address general concerns below, followed by specific comments.

**General Comments**

The main contribution of this work is to develop a framework to propagate uncertainties in ocean melt forcing (originating from a large range of sources) to uncertainties in the Antarctic mass loss. I am glad to see that this focus is highlighted by the authors throughout the manuscript. For developing such framework, however, more analysis is needed, in particular regarding the limitations of the currently proposed approach. This should include, as far as possible, a benchmark of the approach to successfully represent uncertainties.

For the currently proposed approach I am concerned about the influence of the input space boundaries on the analysis and the unclear (to me) measures taken to elevate this. Several of the optimal fitting parameters for t0 and (maybe) tau appear to be at the edge or even outside of the considered input

space. As the authors note, this is a problem because large parts of derived priors lie outside of the considered space which cannot be (well) represented in the following analysis. The authors shift the fitted parameter values away from these boundaries (to a value 'within 1 sigma of best fit') which is, however, no appropriate measure to handle this limitation. First, it is not clear what data the sigma (=standard deviation?) is based on and which/how many parameter values are effected. But more importantly, by (seemingly arbitrarily) changing the values you are not representing the ocean model melt forcing anymore which undermines the following analysis.

The preferred solution would be to run additional CISM simulations to expand the input space. In case they exist, you could also give strong arguments why such simulations would have to be considered unrealistic (a-priory) to justify the prior cut-off. Otherwise you would, at the very least, have to show the impact of said boundary cut-offs on the sea level rise distribution (see quality assessment below).

The second concern are the sigmoid parameter distributions themselves (Figure 10). In some cases you have as little as two values (Peninsula) to fit a Gaussian distribution to, which leads without doubt to a large standard error of the mean. In addition you treat all sigmoid parameters as statistically independent. Let me give an example why this is problematic: The sigmoids with [t0=100, tau=70, M200=50] and [t0=0, tau=700, M200=250] result in very similar forms, despite having very different parameter values (I realize that said parameter values are outside of the parameter space you consider, but this is just an example for the existence of ambiguities). Another example is the case where the melt rate is constant in time (M200=0) in which case the other two parameters are completely unconstrained. By an independent treatment you cannot take this kind of ambiguities into account. This raises the question of how the proposed approach to find priors compares to approaches which do not rely on such an assumption of independence. These kinds of limitations are clearly stated by the authors (e.g. line 341-347), but should be addressed by further analysis (i.e. comparing and testing alternative approaches).

Two possible alternative approaches come to mind: (1) The sigmoid parameter priors could be base on the cumulative ocean melt instead of the fitted parameter values. Or (2) the sigmoid parameter distributions could be derived jointly by agreement with the time-series themselves; One possible approach to this (but by no way the only possibility) would be to take the mean and sd. for each year of the ocean model time series for each basin (gray lines in Figure 5) and search the input space for all sigmoids which are in reasonable agreement with this. 'Reasonable agreement' can by defined with a threshold or weighting on any kind of score function (such as a simple RMSE), or possibly by ruling out all sigmoids which are outside of the mean+-3sigma interval for more then 5% of the time (Pukelsheim, Friedrich. "The three sigma rule." The American Statistician 48.2 (1994): 88-91.) and uniform (equally likely) treatment of all passing sigmoids.

In any case, we need to address the quality of the resulting distributions. One easy step of validation would be to inspect the consistency of SLR contributions from the updated emulator priors (i.e. solid lines in Figure 12) with results from the optimal sigmoids (without shifting them, i.e. the ones shown in Figure 5). If there are biases, can they be explained? How compares the spread within the emulated ensemble with the optimal? Similar tests could be done for subsets

(selected or random) of CISM simulations which are treated as synthetic test melt predictions (perfect model test).

As you point out, the very limited number of ocean model melt predictions limits the ability to constrain the prior distributions. Considering the focus on designing and testing statistical methods I would suggest to distinguish between 100-year and 200-year predictions more strongly. For the 100-year period more ocean melt projections are available from the current sources, but also projects like ISMIP6, which would allow for more analysis of the priors (as described above). If appropriate, inter-GCM comparisons could be another option.

**Specific Comments**

**L1:** 'and have already begun thinning in response to increased basal melt rates' Many Antarctic ice shelves are not currently thinning

**L32:** Consider more appropriate reference (e.g. Chapter 4 of IPCC Special Report on the Ocean and Cryosphere in a Changing Climate, or references therein)

**L44 or elsewhere:** consider Holland et al. (2019) for more context (https://doi.org/10.1038/s41561-019-0420-9)

**L40:** -'the forefront' +'a focus area'

**L41:** comma after 'those that do'

**L70-74:** This is a good explanation of emulators

**L95:** here and elsewhere: ice model -> ice sheet model

**L109:** Since you combine (2) and (3) of the list above, I am skeptical about 'is intended to be agnostic with respect to these assumptions'. Maybe something like: 'is intended to overcome some of these limitations'?

**L110:** highly -> densely

**L113:** 'any scientific source': can you be more precise?

**L143:** please specify the SMB forcing here; Is the SMB constant through the spin-up and future? Which years is it based on and how?

**Figure 1:** The order of tasks is a bit confusing; Task 3 is (emulator) validation and Task 5 is emulation. Can Task 5 and 6 be combined?

**L146:** In Seroussi et al. 2020 a 30 000 year spin-up is used. What was the reason to use 40 000 years here and how big is the difference to the ISMIP6 initial state?

**Figure 2:** A difference plot between the modelled and observed speeds would be helpful here, e.g. replacing the observed speed map similar to Figure 3

**L154:** The model is in a steady state and the observations indicate otherwise, this alone can be seen as a contradiction to 'excellent agreement between observed and modeled...', please rephrase.

**L156:** 'very little drift' can you give a number/order of magnitude here?

**L159:** I agree with the editor that the interpretation as preindustrial initial state (after nudging to modern velocities and using ocean melt forcing from 2000 onward) is not helpful, nor necessary.

**L165-L173:** Please clarify: Are any ocean melt predictions used here based on ECHam5 simulations? ECHam5 based ocean melt simulations in both references (Timmermann and Hellmer (2013) and Cornford et al. (2015)) seem to end in year 2100 while all melt forcings shown here are 200 years long.

Please further clarify: Are any ocean melt predictions used here based on the E1 climate scenario? And if so, which? Comparing Figure 4 in Cornford 2015 with Figure 5 in this manuscript makes me wonder why, e.g. the Filcher-Ronne ice shelf time-series for HadCM3 / E1 with BRIOS ocean model is not included as Weddell sea ocean forcing. If there has been a selection process, what is the criterion for selection? A Table with source, ocean model, climate model and climate scenario would help with a more transparent database.

Which are the four East Antarctic Ice Sheet ocean melt predictions used for Figure 5? I am just asking because Timmermann and Hellmer (2013) seem to focus on only three types of 200 year simulations (HadCM3-A1B+FESOM, HadCM3-E1+FESOM and HadCM3-A1B+BRIOS) and Cornford et al. (2015) do not include the EAIS at all. On top of this there seems to be an unfortunate similarity between the highest EAIS melt prediction in Figure 5 and one of the Cornford et al (2015) predictions for the Marie-Byrd land (Figure 4, lower right panel, HadCM3/A1B+Brios). Again, the before mentioned table could help to clarify this.

**L191-193:** The comparison of quasi-random Sobol sequences to pseudo-random Monte Carlo sampling is unclear. What is the difference between quasi-random and pseudo-random? What do you mean by Monte Carlo sampling? Consider here the wide definition of 'Monte Carlo methods' on one side and the popular Monte Carlo Marcov Chains on the other side (which are designed to build clusters to represent priors). This is probably also a good place to mention that the Sobol sequence has a uniform distribution though the input parameter space (unless it is already mentioned in this section somewhere?).

**L199-201:** 'capturing all of the sigmoidal characteristics seen in the modeled ocean melt rate projections'; This creates the impression that the parameter ranges include all optimal sigmoidal fits to ocean model melt projections. Some of the lines in Figure 5 appear to have t0 values outside of the 100-225 year range. Please clarify whether this is the case and consider this in the impact/justification discussion of input space boundaries.

**L224:** Is the melt anomaly also imposed at newly ungrounding locations?

**L231:** 'highly sample' -> 'densely sample'

**Figure 5 and elsewhere:** m/yr vs. m/y (Figure 11) vs. m/a vs. m a$^{-1}$

**Figure 6 caption:** maybe: '...up to the maximal melt values given in the text, representing roughly twice the maximum 'data' melt rate at year 200'

**Figure 8:** Define the error bars and preferably avoid calling the CISM SLR values 'True' (since it is not the true future SLR contribution). You could use e.g. 'ice sheet model vs. emulated' or 'CISM vs. CISM emulator', etc.

**L249:** This chapter does a good job in explaining Gaussian Process emulation to non-expert audience. However, it is lacking one or two sentences with technical details about the emulator to be reproducible. E.g. 'Usually the smoothness of data being fit is estimated as part of the interpolating procedure.': is this the case here? Using a marginal likelihood optimisation? What covariance function is used? Is the nugget (representing noise in the training data) set to zero since you emulate a deterministic model? What emulator sample size is used?

**L252:** 'have excellent accuracy', in principal emulator and ISM values can be highly correlated but not along the main diagonal, in which case the emulator would not be accurate. Maybe just rephrase to 'show a good emulator performance' or similar.

**L265:** 'corresponding [+rate of] mass loss (Gt/yr)'? please clarify whether this is the rate of mass loss contributing to sea level rise (i.e. from grounded areas), total loss of ice (i.e. including ice shelves) or flux across the grounding line (i.e. dynamic contribution to grounded ice mass balance, which could be balanced by SMB). The same applies to Figure 9, labels and caption. Also consider using mm sea level equivalent instead of Gt for comparability.

**L268:** It is good that you are very clear about this.

**L275-276:** Again, it is not clear how exactly this is done but if I understand it correctly I do not see any good justification for this.

**L277:** What is the impact of using normal distributions? The sample size is probably not large enough (yet) to support or dismiss this assumption, but you could try other shapes of distributions to investigate the sensitivity.

**L280 from 'This falls within...':** consider moving this and/or the remainder of the section to the discussion.

**L291:** Please briefly set this work into relation to Levermann et al. (2020) (LARMIP-2).

**Figure 9:** Restricting the x-axis to 0-200 years would make this (and other) figure look tighter. The yellow bar in the top figure starts at 0, is that correct?

**Figure 9 caption:** blue shading (needs to be better defined) and red curve are introduced twice (for each panel) which might be possible to be combined. (maybe something like: Figure 9. 500-member CISM ensemble during 200-year simulations with mean (red curve) and total range of simulations[?] (blue shading). Shown are the SLR contribution (top) and rate of SLR contribution[?] (bottom)...).

**L292:** Agreed. I hope my remarks will help to deepen the exploration.

**Figure 11:** I assume the differences in 'best fits' to Figure 5 come from the one sigma shifting of the parameters alone? It illustrates well that the priors you generate encompass the 'best fits' quite well, which could be mentioned in the text.

**Figure 12:** This figure shows predominantly the combined effect of constraining the ocean melt forcing and the methodological choices to derive the priors, compared to the training ensemble. As mentioned in the general remarks it should also include the results of the optimal parameter fit simulations/emulations to illustrate the impact of methodological choices (on the priors) alone, as well as new SLR distributions based on additional prior(s).

**L299:** 'we can use it to [+densely] sample'

**L309:** 'the [+emulated] CISM model projects' (or, in case these numbers are not based on the emulator, make this more clear and compare to respective emulated values)

**L460:** The final version of this reference has become available

**L485:** The final version of this reference has become available

---

## Referee Comment (RC2) · Anonymous Referee #2 · 20 Sep 2020

**Review Report on TC-2020-178: "Statistical emulation of a perturbed basal melt ensemble of an ice sheet model to better quantify Antarctic sea level rise uncertainties"**

The manuscript proposes a new way to build a statistical surrogate for an expensive Antarctic ice sheet model which can quickly generate future SLR values given a basal melt rate anomaly trajectory, which represents the trajectory of sub-shelf ocean forcing. The proposed approach parameterizes the anomaly trajectories using sigmoid curves and builds a Gaussian process emulator for the relationship between the parameters of the sigmoid curves and the SLR values in the target year. This paper addresses the problem of accounting for ocean forcing uncertainties in future Antarctic ice sheet projections, one of the long-standing issues in the ice model community, and hence has scientific merit that warrants publication in *the Cryosphere*. However, I think the following major and minor points that I list below need to be addressed or at least seriously discussed in the manuscript before being considered for publication.

Major Points

1. I am mainly concerned about how realistic the smoothed the basal melt rate anomaly trajectories are and, if not, how the unrealistic (perhaps oversmoothed) forcing trajectories affect the realism of the final SLR projections. For example, while the authors argue that the basal melt rate anomaly trajectories from Timmermann and Hellmer (2013) and Cornford et al. (2015) in Figure 5 are accurately captured by the sigmoid functions, I can see that a lot of mid-range temporal patterns are smoothed out. For example, in the second (Ross Island) panel there are some clear discrepancies between the fitted sigmoid curves and the original trajectories and the fitted sigmoid curves clearly underestimate the basal melt rate anomalies in the end. Will the overly smoothed trajectories lead to vastly different SLR distributions compared to the unsmoothed trajectories? My worry is that using smoothed forcing trajectories may result in notably smaller SLR variations (as the resulting simulated SLR trajectories might be also overly smoothed) than the variations that would have been obtained without smoothing the basal melting rate anomalies. One easy way to check if this is the case is to obtain a few ice sheet model runs using the original basal melting rate anomalies from Timmermann and Hellmer (2013) and Cornford et al. (2015) and see how the final results differ from the runs based on the smoothed trajectories.

2. If the smoothing indeed leads to underestimation of the SLR uncertainties, one way to solve the issue might be to add some additional noise generated from temporally dependent processes such as the ARMA model to the simulated SLR trajectories. The parameters for the ARMA model might be estimated by comparing the SLR projections generated based on the original basal melting rate anomalies and those generated based on the corresponding sigmoid curves.

3. In Lines 273-275, the authors mention that 'least-squares optimization' is done to find the best fit. However I cannot find what variables are actually used in the 'least-squares optimization' here. Are they the simulated SLR trajectories and some observational data? Or are they the fitted sigmoid curves and the original basal melting rate anomalies? Judging based on the caption in Figure 6, I think it is the latter. Then I think the issue can be easily solved by expanding the plausible ranges and also running more ice sheet model runs and obtaining more emulated runs accordingly so that the envelop of the colored curves shown in Figure 6 well-contain the black curves. I am not sure why the authors are relying on some ad-hoc procedure to fix the issue instead of expanding the plausible ranges.

Minor Points

1. Related to the major point #1 above, there is an existing method to emulate the future projections for different forcing scenarios (Catruccio et al. 2014). I think it will be ideal to compare the proposed method with this approach, but it might require too much effort to repurpose this method for ice sheet projection. I will leave the decision to the authors, but I think it is at least worth mentioning this approach as a possible future direction.

2. The authors use Kennedy and O'Hagan (2001) as the main reference for Gaussian process-based emulation, but that idea should be attributed to Sacks et al. (1989). In fact the main contribution of Kennedy and O'Hagan (2001) is more on the calibration side rather than the emulation side.

3. Related to the major point #3 above, having estimated parameter values that are at or outside of the plausible parameter ranges for a model ensemble is a well-known issue in computer model calibration literature (see. e.g., Brynjarsdóttir and O'Hagan,2014, Chang et al., 2016, Salter et al., 2019). In fact, this is a typical example of a 'terminal case' mentioned in Salter et al. (2019).

References

Chang, W., Haran, M., Applegate, P.J., and Pollard, D. (2016) Improving ice sheet model calibration using paleoclimate and modern data, *the Annals of Applied Statistics*, 10 (4), 2274-2302

Jenný Brynjarsdóttir and Anthony O'Hagan 2014 *Inverse Problems* 30 114007

Sacks, Jerome; Welch, William J.; Mitchell, Toby J.; Wynn, Henry P. Design and Analysis of Computer Experiments. *Statistical Science* 4 (1989), no. 4, 409-423. doi:10.1214/ss/1177012413. https://projecteuclid.org/euclid.ss/1177012413

Castruccio, Stefano, David J. McInerney, Michael L. Stein, Feifei Liu Crouch, Robert L. Jacob, and Elisabeth J. Moyer. "Statistical emulation of climate model projections based on precomputed GCM runs." *Journal of Climate* 27, no. 5 (2014): 1829-1844. (website: http://www.rdcep.org/research-projects/climate-emulator, source code: https://github.com/RDCEP/climate_emulator)

Salter, M. J., Daniel B. Williamson, John Scinocca & Viatcheslav Kharin (2019) Uncertainty Quantification for Computer Models With Spatial Output Using Calibration-Optimal Bases, *Journal of the American Statistical Association*, 114:528, 1800-1814

---

## Author Comment (AC1) · 13 Mar 2021

**Responses to Reviewer 1**

**Reviewer response to the manuscript: 'Statistical emulation of a perturbed basal melt ensemble of an ice sheet model to better quantify Antarctic sea level rise uncertainties'**

Anonymous

September 4, 2020

We thank the reviewer for their very thorough and insightful review of this paper. We believe the changes made in response to these comments make the manuscript stronger overall. Below, comments in blue are our direct responses to the reviewer's comments.

Based on a large ensemble of the state-of-the-art Community Ice Sheet Model simulations the authors develop a flexible and efficient method to represent the Antarctic ice sheet under a wide range of ocean melt forcings, a major part of the general climate forcing in Antarctica. The manuscript explores the use of an statistical emulator for updated projections of 200-year sea level rise contributions. The link between global climate predictions and the Antarctic response is a key challenge the ice sheet modelling community is currently facing and the use of advanced uncertainty quantification methods, as done here, is likely to play an important role for this.

The presented work is therefore of very high relevance, however, I have concerns regarding the proposed approach to find probabilistic projections and a lack of testing of this approach. These are largely evolving around refining the parameter priors which should make it easier to address my concerns for re-vision. Once this is done I see the potential for follow up studies to combine multi-GCM southern ocean climate prediction distributions with ocean melt prediction PDFs and the ice sheet model sea level emulator developed here. This could be an major step towards a quantification of the sea level rise projection uncertainties. I will address general concerns below, followed by specific comments.

General Comments

The main contribution of this work is to develop a framework to propagate uncertainties in ocean melt forcing (originating from a large range of sources) to uncertainties in the Antarctic mass loss. I am glad to see that this focus is highlighted by the authors throughout the manuscript. For developing such framework, however, more analysis is needed, in particular regarding the limitations of the currently proposed approach. This should include, as far as possible, a benchmark of the approach to successfully represent uncertainties.

For the currently proposed approach I am concerned about the influence of the input space boundaries on the analysis and the unclear (to me) measures taken to alleviate this. Several of the optimal fitting parameters for t0 and (maybe) tau appear to be at the edge or even outside of the considered input space. As the authors note, this is a problem because large parts of derived priors lie outside of the considered space which cannot be (well) represented in the following analysis. The authors shift the fitted parameter values away from these boundaries (to a value 'within 1 sigma of best fit') which is, however, no appropriate measure to handle this limitation. First, it is not clear what data the sigma (=standard deviation?) is based on and which/how many parameter values are effected. But more importantly, by (seemingly arbitrarily) changing the values you are not representing the ocean model melt forcing anymore which undermines the following analysis.

We agree with the reviewer here regarding the lack of clarity around the method to shift parameters away from the bounds. We no longer include such a method, but for clarification for the reviewer we feel compelled to explain more clearly how this shifting was implemented, and why it was done.

First, we want to clarify the actual method used for moving away from the edge of the parameter space. The parameter values themselves were not shifted by 1 sigma (standard deviation), but instead we relaxed the 'best fit sigmoid' criteria such that the best fit could occur anywhere within a 1 sigma window around the optimally fit sigmoid. In this way we were able to find a sigmoid fit within this window that was characteristic of the ocean model output but was not using edge-hitting parameters. The parameter ranges were originally chosen based on expert judgement (though in one case we caveat this turned out to be too narrow, and we discuss this in further detail below).

We believe the boundary-hitting behavior we see is evidence of non-identifiability (compensating parameter errors) rather than that the ranges are too narrow. The reviewer notes this same issue of non-identifiability later in the review, and we agree with their assessment. Indeed, edge-hitting parameter estimates do not imply that the 'best fit' curves lie outside of our range. Hitting edges can occur due to confounding between parameters, meaning there are many ways of generating parameter combinations for equally good fits (both inside and outside the parameter space). When there are identifiability issues due to confounding between parameters, we generally expect (nearly) equally-good parameter fits to lie on some "ridge" or low-dimensional manifold within parameter space. As a consequence, we can find "good fits" for unphysical parameter values. Therefore, we tried our best to cut off the unphysical range with our prior.

We illustrate this issue further by showing examples of equally good fits within and outside of our parameter ranges. Fig R1.1 shows this for two ocean model projections. In the case of the Ross region (left panel), a similar fit can be found by increasing both t0 and Mmax. As t0 (the inflection point) is pushed later, Mmax also must increase to compensate. For the Weddell region (right panel) a similar fit can be found if tau is increased. In that case M increases so that the curve grows to a higher value but more slowly.

[Figure]

**Fig R1.1:** Ocean model melt rate time series (grey) as well as the best fit in our bounds (blue) and out of our bounds (orange) for the Ross region (left) and Weddell region (right). We see in the left panel that the t0 value in the blue curve is hitting the upper edge of our boundary (t0=225). The RMSE for in bounds = 0.74, RMSE for out of bounds = 0.71. For the Weddell (right), t0 is edge-hitting on the lower boundary (t0=100). By increasing tau, t0 and M also must increase to compensate. The RMSE for the curve in our bounds = 0.47 and RMSE for the curve out of bounds = 0.48.

Overall, we believe the existence of edge-hitting parameter values does not imply in most cases that our ranges are too small, because equally good fits occur within our ranges, and thus do not believe that expanding the ranges will necessarily achieve better fits.

Therefore, our rationale for the ad-hoc methodology was to provide a better basis for a multi-model distribution over the point estimates obtained from fitting the sigmoid curve to different ocean models. If many parameter settings provide equally good fits due to non-identifiability, we could safely seek a 'central' value for each model parameter while maintaining a good fit to the data. This could then be used as a point estimate from which to construct a multi-model normal distribution, avoiding a heavily truncated distribution.

We found that, for the most part, edge-hitting was not a problem for this reason of non-identifiability. However, in the case of one ocean model (BRIOS ocean model), we have evidence that none of the parameters within our prior range provide good fits (Fig R1.2). The physical origin of the misfit is that our range did not allow for earlier t0 values that correspond to earlier inflection points in the curve. As a result we no longer include this model, and only focus on the FESOM model to test our emulator. We will caveat this in the text.

[Figure]

**Fig R1.2:** Ocean melt rate anomalies from the BRIOS ocean model output (red), best fit sigmoid outside of our bounds (green) and best fit inside of our bounds (blue).

All in all, we agree with the reviewer that our previous methodology of shifting away from the edges was still left wanting. We no longer include this procedure and have instead replaced it with three new methods of generating prior distributions. These are described in more detail below.

The preferred solution would be to run additional CISM simulations to expand the input space. In case they exist, you could also give strong arguments why such simulations would have to be considered unrealistic (a-priory) to justify the prior cut-off. Otherwise you would, at the very least, have to show the impact of said boundary cut-offs on the sea level rise distribution (see quality assessment below).

Unfortunately, we cannot run any more simulations as we no longer have computational time available for use. Available models were examined at the time the ensembles were designed in order to decide on prior cut-offs. As stated in our response to the previous comment, the ranges were decided primarily on the fits to the available data at the time and expert judgement. Since it is possible to find "good fits" for unphysical parameter values, we tried our best to cut off the unphysical range. We've articulated these limitations in the revised text.

The second concern are the sigmoid parameter distributions themselves (Figure 10). In some cases you have as little as two values (Peninsula) to fit a Gaussian distribution to, which leads without doubt to a large standard error of the mean. In addition you treat all sigmoid parameters as statistically independent.

Let me give an example why this is problematic: The sigmoids with [t0=100, tau=70, M200=50] and [t0=0, tau=700, M200=250] result in very similar forms, despite having very different parameter values (I realize that said parameter values are outside of the parameter space you consider, but this is just an example for the existence of ambiguities). Another example is the case where the melt rate is constant in time (M200=0) in which case the other two parameters are completely unconstrained. By an independent treatment you cannot take this kind of ambiguities into account. This raises the question of how the proposed approach to find priors compares to approaches which do not rely on such an assumption of independence. These kinds of limitations are clearly stated by the authors (e.g. line 341-347), but should be addressed by further analysis (i.e. comparing and testing alternative approaches).

We now include three different methods of extracting prior estimates aimed at this ambiguity issue by considering a windowed set of 'good fits' (as the reviewer suggests) rather than a single point estimate. The methods are described in detail below.

- **Method 1: Individual fits + normal distribution**

This method is performed by generating a distribution of prior parameters based on the 'best fits'. The best fits are found by a least-squares optimization between the fitted sigmoid curves and the original basal melting rate anomalies from the ocean models (Fig. 5 colored and grey curves respectively). The sigmoid parameters that describe the data fits are then used to generate a normal distribution that serves as the prior. This final step is to allow for the possibility that other ocean models, not considered here, could lead to plausible parameter values. The emulator then samples parameters from this distribution. These prior distributions can be combined and presented as a distribution of sigmoid anomalies (Fig. 10).

- **Method 2: Window fits + direct sampling**

This method constructs a windowed set of good parameter values for each ocean model. The window size is defined as 2 SD around the best fit sigmoid. Instead of finding a singular best fit to the ocean model as in Method 1 over which a normal distribution is generated, only fits within this window are used. The windowing is to allow for relaxation away from the edge-hitting parameters. For each region there is an equal-probability mixture of 'windowed fits' across the ocean models to represent the multi-model uncertainty. This method does not account for the possibility of melt trajectories not represented by the ocean models.

- **Method 3a: Mixture method (Window fits +normal distribution)**

This method uses a mixture of Methods 1 and 2, an attempt to get the `best of both worlds': account for non-identifiability/ambiguity in model fits by including a windowed set of good fits as in Method 2, but fit a continuous distribution of the model fits so that probability does not concentrate only on the parameter space locations of the ocean models. This gives nonzero probability to ocean melt trajectories that don't come from the ocean models in order to account for multi-model uncertainty. So, the same windowing technique is used as in Method 2, but instead of using the parameters of the windowed curves directly as our priors, we generate a normal distribution around the windowed fits as in Method 1. This may be thought of as an approximation to the hierarchical Bayesian approach taken in Jonko et al. (2018), where the parameters arising from fitting each climate model are assumed to be a sample from an underlying multi-model distribution.

- **Method 3b: Mixture method (Window fits + multivariate normal distribution)**

This method is the same as Method 3a in that we want to allow for the possibility of other ocean models not contained here. However, unlike in method 3a, it does not assume an independent normal distribution for each parameter. Instead, in order to account for correlation across parameters, we use a multivariate normal distribution (aka a tilted normal).

By using these three methods we are trying to account for issues like non-identifiability (by windowing) and multi-model uncertainty (by adding a normal distribution). Our new results therefore include the SLR projections for each of the methods (Fig R1.3 below).

[Figure]

**FigR1.3**: Sea level rise probability distributions for (a) year 100 and (b) year 200. The ensemble SLR PDF at year 100 and 200 (grey), and the predicted SLR PDFs for three prior methodologies described: Individual fits + normal distribution (blue), window fits + direct sampling (orange), window fits + normal distribution (green) and window fits + multivariate normal distribution (red).

This result shows that in both year 100 and 200, the likeliest predicted SLR in both of these years is not strongly dependent on the prior method choice. The addition of a normal distribution to account for multi-model uncertainty (ie. the possibility of other models occupying nearby but not the same parameter space) generally causes the mode of the distribution to shift to slightly larger SLR values. Prior methods 2 (window fits + direct sampling) and 3b (window fits + multivariate normal distribution) produce the lowest SLR prediction for year 100. As expected, by using a multivariate normal (method 3b) instead of a normal distribution (method 3a), the SLR prediction shifts closer to the direct sampling (method 2) prediction which also implicitly has correlations in it. A notable difference in method 2 and method 3b, however, is that method 2 results in bimodality in year 200. This is an artifact of sampling over a small discrete set of ocean models. There is a bimodality for the same reasons in the year 100 prediction for this method as well but is smoothed out when emulator uncertainty is accounted for. Our preferred method is 3b, because it is the most principled approach. These results and associated discussion is included in the text now.

Two possible alternative approaches come to mind: (1) The sigmoid parameter priors could be base on the cumulative ocean melt instead of the fitted parameter values. Or (2) the sigmoid parameter distributions could be derived jointly by agreement with the time-series themselves; One possible approach to this (but by no way the only possibility) would be to take the mean and sd. for each year of the ocean model time series for each basin (gray lines in Figure 5) and search the input space for all sigmoids which are in reasonable agreement with this. 'Reasonable agreement' can by defined with a threshold or weighting on any kind of score function (such as a simple RMSE), or possibly by ruling out all sigmoids which are outside of the mean+-3sigma interval for more then 5% of the time (Pukelsheim, Friedrich. "The three sigma rule." The American Statistician 48.2 (1994): 88-91.) and uniform (equally likely) treatment of all passing sigmoids.

As described above, we now use 3 new methods, two of which include a windowing technique similar (but more conservative) to what the reviewer describes here.

In any case, we need to address the quality of the resulting distributions. One easy step of validation would be to inspect the consistency of SLR contributions from the updated emulator priors (i.e. solid lines in Figure 12) with results from the optimal sigmoids (without shifting them, i.e. the ones shown in Figure 5). If there are biases, can they be explained? How compares the spread within the emulated ensemble with the optimal? Similar tests could be done for subsets (selected or random) of CISM simulations which are treated as synthetic test melt predictions (perfect model test).

This has been addressed above as well.

As you point out, the very limited number of ocean model melt predictions limits the ability to constrain the prior distributions. Considering the focus on designing and testing statistical methods I would suggest to distinguish between 100-year and 200-year predictions more strongly. For the 100-year period more ocean melt projections are available from the current sources, but also projects like ISMIP6, which would allow for more analysis of the priors (as described above). If appropriate, inter-GCM comparisons could be another option.

We agree that considering the ISMIP6 100 year predictions would be interesting, however we do not think it would add much to the current form of the manuscript, which is introducing a technical approach. More ocean models would certainly give a more realistic prior, but our results are anyway not a realistic projection of SLR. Rather, demonstration of our technique is the goal here. In the future, we plan to include the ISMIP6 data after addressing, for example, the equilibrium spin-up issue.

Specific Comments

L1: 'and have already begun thinning in response to increased basal melt rates' Many Antarctic ice shelves are not currently thinning

This has been changed to scale back the statement. It now reads: "Antarctic ice shelves are vulnerable to warming ocean temperatures, and some have already begun thinning in response to increased basal melt rates."

L32: Consider more appropriate reference (e.g. Chapter 4 of IPCC Special Report on the Ocean and Cryosphere in a Changing Climate, or references therein)

Good suggestion, a new reference has been included: "Despite its potential to contribute to sea level rise (SLR) vastly more than any other single source (~5m West Antarctica, ~60 m all Antarctica), and documented ice shelf thinning (e.g. Schroeder et al., 2019; Reese et al., 2018), Antarctica's contribution to sea level remains highly uncertain (Oppenheimer et al., 2019; Heal and Milner, 2014)."

L44 or elsewhere: consider Holland et al. (2019) for more context (https://doi.org/10.1038/s41561- 019-0420-9)

This paper has now been included in our introductory remarks of the paper.

L40: -'the forefront' +'a focus area'

The new sentence reads: "Development of these modeling capabilities is still a major focus area of current research."

L41: comma after 'those that do' Added. Thanks.

L70-74: This is a good explanation of emulators. Thanks.

L95: here and elsewhere: ice model -> ice sheet model  Thanks, these have been updated.

L109: Since you combine (2) and (3) of the list above, I am skeptical about 'is intended to be agnostic with respect to these assumptions'. Maybe something like: 'is intended to overcome some of these limitations'?

Yes, good suggestion.

L110: highly -> densely Done

L113: 'any scientific source': can you be more precise?   This has been updated for specificity. It now reads: "After constructing a statistical emulator of this ensemble, we can then provide the emulator with basal melt assumptions derived from a number of ocean/climate model combinations."

L143: please specify the SMB forcing here; Is the SMB constant through the spin-up and future? Which years is it based on and how?
Yes the SMB is constant for the spin-up and uses a 1976-2016 climatology from RACMO2, and is also held constant for the forward runs.  The following text has been added for clarification:

"Surface mass balance (SMB) from late 20$^{th}$ century simulations with the RACMO2 regional climate model (van Wessem et al., 2018). SMB is held constant using the RACMO2 1976-2016 climatology in the spin-up and forward runs."

Figure 1: The order of tasks is a bit confusing; Task 3 is (emulator) validation and Task 5 is emulation. Can Task 5 and 6 be combined?

The figure has been updated to try and clarify Task 3, as well as combining Task5/6. See the new version below.

[Figure]

L146: In Seroussi et al. 2020 a 30 000 year spin-up is used. What was the reason to use 40 000 years here and how big is the difference to the ISMIP6 initial state?

The spin-up used in our paper was identical to that used in Seroussi et al. (2020) but section C7 in Seroussi has been confirmed to contain a typo. It states the spin-up was 30 000 years, when in fact it was 40 000 years.

More generally, it is worth noting that it doesn't much matter whether you spin up for 30 ka or 40 ka. There is a long period of slow drift as the temperature equilibrates, and results won't be sensitive to a few thousand years of this slow drift.

Figure 2: A difference plot between the modelled and observed speeds would be helpful here, e.g. replacing the observed speed map similar to Figure 3.

We have updated this Figure (below) so that it now includes a panel showing the difference in observed and modeled speeds. Furthermore, the color scheme has been updated to match those in the CISM ISMIP6 paper for consistency. Figure 3 has also been updated for consistency in figure style.

[Figure]

L154: The model is in a steady state and the observations indicate otherwise, this alone can be seen as a contradiction to 'excellent agreement between observed and modeled...', please rephrase.

This is a good point, and the text has been updated for clarity: "The model is run on a uniform 4 km grid, resulting in a spun-up state with good agreement between observed and modeled surface velocity (Fig. 2), ice shelf extent, and ice thickness (Fig. 3), except in regions that are known to be out of steady state, such as the Amundsen sector and the Kamb Ice Stream."

L156: 'very little drift' can you give a number/order of magnitude here?

Yes, this is a good point. The text has been updated to include a quantification of the drift: "A control run starting from the end of spin-up and going forward 1000 yr (not shown) indicates that there is very little drift (<1 Gt/yr) in the ice sheet moving forward and that most changes in ice thickness will be a result of forcing as opposed to internal variability or model drift."

L159: I agree with the editor that the interpretation as preindustrial initial state (after nudging to modern velocities and using ocean melt forcing from 2000 onward) is not helpful, nor necessary.

We agree and have removed the comment that this is closer to a PI state. The text now reads: "A control run starting from the end of spin-up and going forward 1000 yr (not shown) indicates that there is very little drift (< 1 Gt/yr) in the ice sheet moving forward and that most changes in ice thickness will be a result of forcing as opposed to internal variability or model drift. This is not fully realistic, since the real ice sheet is never truly in equilibrium with the climate, particularly if current observations are used to tune the model. Therefore, henceforth, we do not explicitly state the year corresponding to SLR projections. Rather, we refer to our SLR projections as relative to the number of years run forward in the model from the end of spin-up. As a result, the sea level rise projections are not tied to a particular year in the future. Rather, they are meant to show that the emulator is a powerful and useful tool, and SLR predictions are considered a proof-of-concept.

L165-L173: Please clarify: Are any ocean melt predictions used here based on ECHam5 simulations? ECHam5 based ocean melt simulations in both references (Timmermann and Hellmer (2013) and Cornford et al. (2015)) seem to end in year 2100 while all melt forcings shown here are 200 years long.

The reviewer is correct, there are no ECHam5 simulations included here because they only go to 2100. We only use ocean simulations that go out to the year 2200. The text has been updated to clarify which

model combinations are used, along with a Table that shows data sources and ocean simulations used in this work.

Please further clarify: Are any ocean melt predictions used here based on the E1 climate scenario? And if so, which? Comparing Figure 4 in Cornford 2015 with Figure 5 in this manuscript makes me wonder why, e.g. the Filcher-Ronne ice shelf time-series for HadCM3 / E1 with BRIOS ocean model is not included as Weddell sea ocean forcing. If there has been a selection process, what is the criterion for selection? A Table with source, ocean model, climate model and climate scenario would help with a more transparent database.

A table has now been added that describes the data sources for each region as well as the model combinations each uses. We are only using the A1B scenario here, and HadCM3 global model. In addition to the table, the text now reads:

We use ocean model data from Timmermann & Hellmer (2013) and Cornford et al. (2015) to inform the types of possible basal melt rate trajectory shapes for 200 years forward. The forcing data for the ocean models are generated with the global climate models HadCM3 (Gorden et al., 2000; Collins et al., 2001) under the A1B emissions scenario. A1B is a moderate scenario similar to Representative Concentration Pathway 6 (RCP6). This is then dynamically downscaled by two high-resolution atmosphere models (RACMO2 and LMDZ4) and two ocean models: the medium resolution BRIOS Bremerhaven Regional Ice-Ocean Simulation (BRIOS) (Timmermann et al., 2002) and the higher resolution Finite-element Sea ice-ocean model (FESOM) (Wang et al., 2014).

Which are the four East Antarctic Ice Sheet ocean melt predictions used for Figure 5? I am just asking because Timmermann and Hellmer (2013) seem to focus on only three types of 200 year simulations (HadCM3- A1B+FESOM, HadCM3-E1+FESOM and HadCM3-A1B+BRIOS) and Cornford et al. (2015) do not include the EAIS at all. On top of this there seems to be an unfortunate similarity between the highest EAIS melt pre- diction in Figure 5 and one of the Cornford et al (2015) predictions for the Marie-Byrd land (Figure 4, lower right panel, HadCM3/A1B+Brios). Again, the before mentioned table could help to clarify this.

As suggested, a Table has now been added and the text has been clarified to reflect the data sources. We only use HadCM3+A1B from both Cornford and Timmermann. The reviewer did catch a typo in the EAIS melt, where we erroneously included the MBL from Cornford (2015) in Figure 5. It is also worth noting that MBL was not included in the SLR predictions, so it was simply an erroneous inclusion in the Figure. This has been rectified.

L191-193: The comparison of quasi-random Sobol sequences to pseudo-random Monte Carlo sampling is unclear. What is the difference between quasi- random and pseudo-random? What do you mean by Monte Carlo sampling? Consider here the wide definition of 'Monte Carlo methods' on one side and the popular Monte Carlo Marcov Chains on the other side (which are designed to build clusters to represent priors). This is probably also a good place to mention that the Sobol sequence has a uniform distribution though the input parameter space (unless it is already mentioned in this section somewhere?).

"Monte Carlo sampling", in this context, refers to independent random sampling from a distribution of parameters. In this case, we are sampling from bounded uniform distributions on each parameter to generate the emulator training ensemble. Computationally, this is often done by pseudorandom sampling, the standard means of generating statistically-random samples using a deterministic numerical algorithm.

Truly random sampling is expected to produce, at random, gaps and clusters within parameter space. To train an emulator, we prefer a more uniform sampling of parameter space. Non-random methods to produce uniform, space-filling sample designs include simple grid (Cartesian product) designs, or the popular Latin hypercube design with better space-filling properties. However, one limitation of the Latin hypercube design is that it is not possible to add new design points (while maintaining the Latin hypercube's uniform space-filling properties) if additional ensemble members are later desired, for example if the original ensemble proves too small or more computer time becomes available.

"Quasi-Monte Carlo (QMC) sampling" provides an alternate means to generate uniform space-filling sampling designs, which allows for the addition of new samples. Like (pseudo)random numbers, QMC produces an arbitrarily long sequence of numbers, and more samples can be taken from this sequence if more ensemble members are desired. However, QMC samples also fill space more uniformly than do true random numbers. (They also have good statistical properties: even though they are not truly random, expectations computed using QMC can converge faster than the 1/sqrt(N) rate of Monte Carlo methods arising from the central limit theorem.) Thus, they can provide a "best of both worlds": the sequence of design points can be extended, like MC sampling, while retaining some of the uniform space-filling properties of non-random designs like Latin hypercube sampling.

We have clarified this in the text, and included the note about the Sobol' sequence having a uniform distribution through the input parameter space.

L199-201: 'capturing all of the sigmoidal characteristics seen in the modeled ocean melt rate projections'; This creates the impression that the parameter ranges include all optimal sigmoidal fits to ocean model melt projections. Some of the lines in Figure 5 appear to have t0 values outside of the 100-225 year range. Please clarify whether this is the case and consider this in the impact/justification discussion of input space boundaries.

This point is discussed earlier in our responses to the reviewer. In short, we did find one model that was not well captured with our parameter space, so it is no longer included. Some new text has been included in the Methods and Discussion to justify our decision of input space boundaries and discuss its limitations.

L224: Is the melt anomaly also imposed at newly ungrounding locations?

Yes this is true, we have added a sentence to make this clear to the reader:

"The melt anomaly is applied to any newly ungrounded cells that appear through the simulation."

L231: 'highly sample' -> 'densely sample'

Done

Figure 5 and elsewhere: m/yr vs. m/y (Figure 11) vs. m/a vs. m a$^{-1}$

Thanks, this has been updated through the text and figure captions to [m/yr] everywhere for consistency.

Figure 6 caption: maybe: '...up to the maximal melt values given in the text, representing roughly twice the maximum 'data' melt rate at year 200'

Done

Figure 8: Define the error bars and preferably avoid calling the CISM SLR values 'True' (since it is not the true future SLR contribution). You could use e.g. 'ice sheet model vs. emulated' or 'CISM vs. CISM emulator', etc.

We agree and have changed the labels here to CISM vs CISM emulator. The error bars represent ±2sigma and that has been noted in the caption here too.

L249: This chapter does a good job in explaining Gaussian Process emulation to non-expert audience. However, it is lacking one or two sentences with technical details about the emulator to be reproducible. E.g. 'Usually the smoothness of data being fit is estimated as part of the interpolating procedure.': is this the case here? Using a marginal likelihood optimisation? What covariance function is used? Is the nugget (representing noise in the training data) set to zero since you emulate a deterministic model? What emulator sample size is used?

We are using the 'GPfit' R package (Macdonald, Ranjan, and Chipman, *JSS* 64, 1, 2015; Ranjan, Haynes, and Karsten, *Technometrics* 53, 366, 2011). We use the standard squared-exponential covariance with independent (factorized) correlation functions for each parameter, and a small nugget for numerical conditioning. The Gaussian process variance hyperparameter is estimated analytically, as is the nugget (following the lower bound given in Ranjan et al., 2011), whereas the (reparameterized) correlation length scale parameters are fit by minimizing the negative profile log-likelihood.

This information is now also included in the text.

We assume the reviewer is asking what the training ensemble size is, which we state in the text to be 500 members.

L252: 'have excellent accuracy', in principal emulator and ISM values can be highly correlated but not along the main diagonal, in which case the emulator would not be accurate. Maybe just rephrase to 'show a good emulator performance' or similar.

The text has been updated now to read: "The predicted SLR values at years 100 and 200 show good emulator performance with correlation coefficients of 0.98 and 0.99, respectively, against the withheld CISM output (Fig 8)."

L265: 'corresponding [+rate of] mass loss (Gt/yr)'? please clarify whether this is the rate of mass loss contributing to sea level rise (i.e. from grounded areas), total loss of ice (i.e. including ice shelves) or flux across the grounding line (i.e. dynamic contribution to grounded ice mass balance, which could be balanced by SMB). The same applies to Figure 9, labels and caption. Also consider using mm sea level equivalent instead of Gt for comparability.

The bottom panel which showed Gt/yr is no longer included, as we decided it did not add to the paper. As such, the discussion of it has also been removed from the text which now reads:

"The CISM ensemble consists of 500 members where each member is forced by five melt rate trajectories, one in each basin. Figure 9 shows the sea level rise time series resulting from the full 500-member ice sheet ensemble (blue shading), with the ensemble mean shown in red. The distributions of SLR at year 100 are more constrained (ranging from 0.5 -- 96 mm) than those in year 200 (ranging from 33 -- 543 mm) (Figure 9 inset). Again, we note that these SLR projections are not physically meaningful

since the parameter sampling over which the ensemble is created is uniform. The ensemble is designed purely to be used for the creation of an emulator."

L268: It is good that you are very clear about this. =)

L275-276: Again, it is not clear how exactly this is done but if I understand it correctly I do not see any good justification for this.

We agree, and so no longer use this method. This statement and all related analysis has been removed and replaced with an explanations of 3 different methodologies we use now. A full description of these methods can be found earlier in this response to the reviewer.

L277: What is the impact of using normal distributions? The sample size is probably not large enough (yet) to support or dismiss this assumption, but you could try other shapes of distributions to investigate the sensitivity.

As part of our three new prior methods, we now include an option with no assumption of a distribution beyond the windowed fits, as well as a normal and multivariate normal distribution. We find the sensitivity to these assumptions to be minor overall, though we do discuss their effect on the SLR predictions in the text now.

L280 from 'This falls within...': consider moving this and/or the remainder of the section to the discussion.

Some of this has been moved to the discussion. The text that emphasizes that these predictions are not tied to a specific year in the future remains, simply because we want to make sure this is clear to the reader when reading the Results section.

L291: Please briefly set this work into relation to Levermann et al. (2020) (LARMIP-2).

This has been done. The new paragraph now reads:

The goal of this work is an in-depth exploration of statistical methods designed to project the effects of a plausible range of sub-shelf ocean forcing conditions upon Antarctic sea level rise uncertainty. We have presented an emulator-based approach to derive probabilistic projections of Antarctic sea level rise from a large perturbed basal melt rate ensemble of ice sheet model simulations. This work comes on the heels of other community efforts to quantify uncertainties in Antarctic sea level rise. For example, the LARMIP-2 project Leverman et al. (2020) applies a linear response theory approach to 16 different ice sheet models (including CISM) in order to estimate the uncertainty of Antarctica's future contribution to global sea level rise that arises from uncertainties in ocean forcing. Their method, similar to that in Castruccio et al. (2014), relies on the assumption of linearity in the ice sheet response, which is generally valid for moderate basal melt rates but tends to break down (including in the CISM model) at higher melt rates, particularly after the first century of simulation. Our emulator method, on the other hand, does not rely on a linearity assumption and is thus valid over a very wide range of ocean scenarios, including the stronger forcing regimes. It is in the high-end (tail-area) ocean forcing scenarios where the greatest societal risk lies, so our focus is to carefully represent those accurately. In the future we could consider a more direct comparison of our results to the linear response approaches used by Levermann et al. (2020).

Figure 9: Restricting the x-axis to 0-200 years would make this (and other) figure look tighter. The yellow bar in the top figure starts at 0, is that correct?

Thanks for the suggestion, the changes to the x-axis on this and other figures has been implemented. The yellow bar in Fig 9 has also been extended down to 0 as the reviewer points out.

Figure 9 caption: blue shading (needs to be better defined) and red curve are introduced twice (for each panel) which might be possible to be combined. (maybe something like: Figure 9. 500-member CISM ensemble during 200- year simulations with mean (red curve) and total range of simulations[?] (blue shading). Shown are the SLR contribution (top) and rate of SLR contribution[?] (bottom)...).

This has been updated. We have also decided that the bottom panel was not necessary, and it has been removed.

L292: Agreed. I hope my remarks will help to deepen the exploration. Great, we hope to have done just that.

Figure 11: I assume the differences in 'best fits' to Figure 5 come from the one sigma shifting of the parameters alone? It illustrates well that the priors you generate encompass the 'best fits' quite well, which could be mentioned in the text.

This is no longer relevant because we do not include a method where the best fits are not shifted away from the parameter bound edges. Our responses above dig into the new methods more completely. We will mention how this figure encompasses the best fits fairly well.

Figure 12: This figure shows predominantly the combined effect of constraining the ocean melt forcing and the methodological choices to derive the priors, compared to the training ensemble. As mentioned in the general remarks it should also include the results of the optimal parameter fit simulations/emulations to illustrate the impact of methodological choices (on the priors) alone, as well as new SLR distributions based on additional prior(s).

Yes indeed, the Figure has now been updated (see FigR1.3). We now include the results for the different methodological choices for prior derivations, along with the training ensemble. We no longer include a method that shifts the priors away from the bounds in an ad-hoc manner.

L299: 'we can use it to [+densely] sample'  Done

L309: 'the [+emulated] CISM model projects' (or, in case these numbers are not based on the emulator, make this more clear and compare to respective emulated values)  Done

L460: The final version of this reference has become available                Done
L485: The final version of this reference has become available                Done

Other references have been updated as necessary.

Responses to Reviewer 2

**Review Report on TC-2020-178: "Statistical emulation of a perturbed basal melt ensemble of an ice sheet model to better quantify Antarctic sea level rise uncertainties"**

We thank the reviewer for their thoughtful and insightful comments on this work. Our responses can be found below inline in blue text.

The manuscript proposes a new way to build a statistical surrogate for an expensive Antarctic ice sheet model which can quickly generate future SLR values given a basal melt rate anomaly trajectory, which represents the trajectory of sub-shelf ocean forcing. The proposed approach parameterizes the anomaly trajectories using sigmoid curves and builds a Gaussian process emulator for the relationship between the parameters of the sigmoid curves and the SLR values in the target year. This paper addresses the problem of accounting for ocean forcing uncertainties in future Antarctic ice sheet projections, one of the long-standing issues in the ice model community, and hence has scientific merit that warrants publication in *the Cryosphere*. However, I think the following major and minor points that I list below need to be addressed or at least seriously discussed in the manuscript before being considered for publication.

Major Points

1. I am mainly concerned about how realistic the smoothed the basal melt rate anomaly trajectories are and, if not, how the unrealistic (perhaps oversmoothed) forcing trajectories affect the realism of the final SLR projections. For example, while the authors argue that the basal melt rate anomaly trajectories from Timmermann and Hellmer (2013) and Cornford et al. (2015) in Figure 5 are accurately captured by the sigmoid functions, I can see that a lot of mid-range temporal patterns are smoothed out. For example, in the second (Ross Island) panel there are some clear discrepancies between the fitted sigmoid curves and the original trajectories and the fitted sigmoid curves clearly underestimate the basal melt rate anomalies in the end. Will the overly smoothed trajectories lead to vastly different SLR distributions compared to the unsmoothed trajectories? My worry is that using smoothed forcing trajectories may result in notably smaller SLR variations (as the resulting simulated SLR trajectories might be also overly smoothed) than the variations that would have been obtained without smoothing the basal melting rate anomalies. One easy way to check if this is the case is to obtain a few ice sheet model runs using the original basal melting rate anomalies from Timmermann and Hellmer (2013) and Cornford et al. (2015) and see how the final results differ from the runs based on the smoothed trajectories.

We appreciate this concern; unfortunately we are unable to run any more simulations in order to explicitly test the effects of the smoothing on the mass loss projections. However, we do not believe that this is necessary to justify our smoothing techniques here. First, we want to reiterate that we are only interested in generating a probabilistic SLR at a specific point in the future (2100, 2200) and we are not attempting to reproduce the transient SLR variability. Furthermore, we physically expect the ice sheet response to be highly integrated (convolved) with respect to basal melt rates, with long timescales, that will tend to eliminate the effects of small rapid stochastic forcings.

Based on previous work by Hoffman et al (2019), Holland (2017), and Robel et al (2019), we do not believe the smoothing will cause vastly different SLR distributions compared to unsmoothed trajectories. (Indeed, one could argue that other important work on Antarctic SLR uncertainty has had to rely on fairly broad assumptions as well (e.g. the assumption of linearity in ice sheet response to forcing in Levermann et al. (2020)). We hope to support this with the following points:

a) Hoffman et al (2019) ran an ice sheet model of Thwaites Glacier, that includes synthetic short-term fluctuations in ocean temperature to investigate if the effect of high-frequency variability on glacier retreat.  They found that for long ice sheet simulations (500 years), fluctuations with periods of 5, 20 and 70 year periods indeed caused a slower retreat. However, the strongest effect they found was up to 10% difference in mass loss at the end of the 500 year run for the longest period of variability (70 yr). For the period of 5 years, the mass loss reduction is only about 2% at year 500. For shorter than 5 year variability, they suggest the effect becomes increasingly trivial. This is because periods below 5 years are too short to support an equilibrium response in ice shelf melting for a small, warm ice shelf such as Thwaites due to incomplete flushing of the sub-shelf cavity (Holland, 2017). **In our case,** most of the variability in the ocean melt rates is 5 years or less, so we would expect minimal effects on mass loss for a 200 year long simulation.

b) The reviewer points out that they are concerned about longer periods of variability (such as that seen in the Ross region (Fig 5 in the paper).  Hoffman et al (2019) show that the influence of all periods and amplitudes of variability do not manifest in large SLR differences (compared to the control) before the year 200 (Fig R2.1, below).  Since our runs are only up to 200 years, it is unlikely that our smoothing will have a large impact on SLR projections.

[Figure]

**FigR2.1** This shows the difference in SLR contribution for each ensemble relative to the control run (Fig 5d in Hoffman et al., 2019). The colors represent the different ensembles of imposed synthetic noise, each with different amplitudes and periods.  This nicely illustrates that SLR slowing as a result of added variability is minimal in the first few centuries of a simulation. The effects really take hold after year 300.  This experiment is just for the Thwaites glacier, so the size and bedrock topography play a large role in how each cavity will individually respond.  However, we suggest this indicates a small effect on shorter time scales.

c)  Hoffman et al. (2019) also evaluate the fractional uncertainty for each of their model ensembles à la Robel et al (2019) and found that at around 200 years, the biggest effect that ocean variability had on total mass loss uncertainty was about 2%. Therefore, in our relatively short simulations of 200 years long, we do not expect the effect of smoothing our basal melt rates to lead to vastly different ice mass loss projections.

Therefore, we do not believe smoothing should cause a large bias in our SLR predictions. **However,** to be sure, we now include additional sources of uncertainty in our emulator. See Point 2 (below) for more details.

2. If the smoothing indeed leads to underestimation of the SLR uncertainties, one way to solve the issue might be to add some additional noise generated from temporally dependent processes such as the ARMA model to the simulated SLR trajectories. The parameters for the ARMA model might be estimated by comparing the SLR projections generated based on the original basal melting rate anomalies and those generated based on the corresponding sigmoid curves.

Thank you for the suggestion here. We can approximate the effects of a stochastic forcing by adding a stochastic term to the response (SLR) in order to account for the over-smoothed forcings (as discussed point 1 above). We added autoregressive noise to the emulated SLR in order to account for natural variability SLR. Furthermore, in order to do a full accounting of uncertainty, we have *also* accounted for code (emulator) error.

We estimate the magnitude of natural variability from the Rignot et al. (2019) mass loss data in Fig R2.2 below. The purple curve shows a time series of the total Antarctic mass loss with error bars in billions of tons. We approximate the standard deviation to be about 500 Gt, or roughly 1.4 mm of SLR equivalent. We note that the natural variability here is not very large compared to the forced secular trend so its impact on SLR predictions will be minimal.

[Figure]

**FigR2.2:** Figure from Rignot et al. (2019). Time series of cumulative anomalies in SMB (blue), ice discharge (D, red), and total mass (M, purple) with error bars in billions of tons for Antarctica, with mean mass loss in billions of tons per year and an acceleration in billions of tons per year per decade for the time period 1979 to 2017. The balance discharge is SMB1979−2008. Note that the total mass change, M = SMB − D, does not depend on SMB1979−2008.

Fig R2.3 below shows a comparison of our SLR predictions with and without the natural variability error included. The solid red curve shows the original SLR distribution, and the blue dotted line shows the effect of when natural variability noise is added. The red dotted line shows when both natural variability and model noise are included. The effect of the natural variability is minor, and in fact in the year 200 it is difficult to make out the difference at all. When both sources of error (natural variability + model noise) are added the effect is larger.

[Figure]

**Fig R2.3**: SLR pdfs for year 100 (left) and 200 (right) showing the difference between the original prediction (solid red), the original prediction + natural variability only (dotted blue), and original prediction + natural variability + model noise (dotted red).

[Figure]

**FigR2.4**: SLR pdfs with and without added both noise terms are shown year 100 (left) and 200 (right). Solid curves show results without noise, and dotted lines show the updated distributions once noise is added.

Fig R2.4 shows the effects of including both additional sources of uncertainty on the three new prior methods discussed below. (A more involved discussion on these three methods can be found above in our response to the reviewer).

The inclusion of noise in the SLR estimates generates a greater effect in year 100 than it does in year 200. We find that the uncertainty from the emulator is generally larger than the uncertainty from natural mass loss variability. Even though the effect of the emulator uncertainty is not large in absolute terms, it can be significant for low SLR values which is why the final SLR distribution is more impacted in year 100 than in 200. This can also be seen in the validation figure (Fig 8 in the paper), which shows that despite a strong correlation between CISM and CISM emulator SLR values, the width of the error bars can be on the same order as the smallest SLR values.

We now present only the 'noise-added' SLR distributions in the manuscript, and we have updated the text to describe how we include natural variability and emulator error into the SLR predictions.

3. In Lines 273-275, the authors mention that 'least-squares optimization' is done to find the best fit. However I cannot find what variables are actually used in the 'least-squares optimization' here. Are they the simulated SLR trajectories and some observational data? Or are they the fitted sigmoid curves and the original basal melting rate anomalies? Judging based on the caption in Figure 6, I think it is the latter. Then I think the issue can be easily solved by expanding the plausible ranges and also running more ice sheet model runs and obtaining more emulated runs accordingly so that the envelop of the colored curves shown in Figure 6 well-contain the black curves. I am not sure why the authors are relying on some ad-hoc procedure to fix the issue instead of expanding the plausible ranges.

First, the reviewer is correct that we are optimizing to fit the sigmoid curves to the regional ocean model basal melt time series. We have clarified this in the text for future readers: "The best fits are found by a least-squares optimization between the fitted sigmoid curves and the original basal melting rate anomalies from the ocean models (Fig. 5 colored and grey curves respectively). The sigmoid parameters that describe the data fits are then used to generate a normal distribution that serves as the prior."

The other reviewer also asked about the possibility of running more simulations. Unfortunately, we no longer have the computer time to run more ensemble members in order to expand the plausible ranges. We believe the boundary-hitting behavior we see is evidence of non-identifiability (compensating parameter errors) rather than that the ranges are too narrow. Edge-hitting parameter estimates do not imply that the 'best fit' curves lie outside of our range. Even if we expanded the parameter ranges, the optimizer may still move along flat ridges of the loss function and hit the boundaries of whatever new ranges we imposed. A lengthier discussion on this, along with evidence of non-identifiability in our fits can be found in the responses to Reviewer 1 (see the first response to General Comments, including figures R1 and R2.)

At the suggestion of the reviewer, we are now recommending three different methods of extracting prior estimates. One of those includes the following procedure: construct a windowed set of good parameter values for each ocean model, construct an (equal) mixture distribution of each of these windowed sets, and then fit a normal & multivariate normal distribution to this mixture of windows to form our prior. The reason we take the last step, instead of using the mixture-of-windows directly, is to allow for the possibility that other ocean models, not considered here, could lead to plausible parameter values not contained within the windows for any of the ocean models. We therefore want to "smooth over" the mixture of windows, to assign nonzero probability to parameter settings that lie near, but not within, the window from any given ocean model. This may be thought of as an approximation to the hierarchical Bayesian approach taken in Jonko et al. (2018), where the parameters arising from fitting each climate model are assumed to be a sample from an underlying multi-model distribution. We include a multivariate normal distribution in order to account for correlation across parameters. This is our preferred method as it is the most principled approach. Again, we refer the reviewer to the responses to Reviewer 1 for more details on all of the new methods that are included and the rationale behind each choice.

Minor Points

1. Related to the major point #1 above, there is an existing method to emulate the future projections for different forcing scenarios (Catruccio et al. 2014). I think it will be ideal to compare the proposed method with this approach, but it might require too much effort to repurpose this method for ice sheet projection. I will leave the decision to the authors, but I think it is at least worth mentioning this approach as a possible future direction.

Based on Eq 1. in Castruccio et al (2014), their method using an "infinite distributed lag model" is similar to what Levermann et al. (2020) did in the LARMIP-2 project. They are estimating the unknown kernel parameter (the decay rate, or response timescale) by inverting a response to a transient forcing (the forcing scenarios shown in Fig. 1 in their paper). Both Castruccio et al. (2014) and Levermann et al. (2020) are building a reduced model based on convolving a response kernel with a forcing. However, Castruccio et al. (2014) parameterized their response kernel as a decaying exponential, whereas Levermann et al. (2020) was able to directly invert for the response kernel nonparametrically from the model response to a step forcing. The reason we did not use such an approach in our work is because of the underlying assumption of linearity, which Levermann showed can break down in some cases (generally in high forcing scenarios after the first century of simulation). However, in the future we might try to compare our emulator to such an approach. We have added text in the Discussion that puts our work in the context of the LARMIP-2 project and also references the relation of Levermann methods to Castruccio's.

Another potential idea for future work might be to recover a response function (convolution kernel) from each ensemble member and use an emulator to interpolate the response functions instead of the actual time series. Effectively, interpolate between response 'models' instead of responses. This may allow for far fewer CISM runs if each linear response model is valid over a wider range of forcings than our current emulator.

2. The authors use Kennedy and O'Hagan (2001) as the main reference for Gaussian process-based emulation, but that idea should be attributed to Sacks et al. (1989). In fact the main contribution of Kennedy and O'Hagan (2001) is more on the calibration side rather than the emulation side.

Thanks for the note on this – the reference has been updated to Sacks et al. (1989).

3. Related to the major point #3 above, having estimated parameter values that are at or outside of the plausible parameter ranges for a model ensemble is a well-known issue in computer model calibration literature (see. e.g., Brynjarsdóttir and O'Hagan,2014, Chang et al., 2016, Salter et al., 2019). In fact, this is a typical example of a 'terminal case' mentioned in Salter et al. (2019).

This is indeed a common issue. In our case, we believe that "edge-hitting" fits are largely is an artifact of non-identifiability between the sigmoid parameters, rather than misspecification / discrepancy of the sigmoid model of basal melt rate trajectories, or too-narrow bounded priors. This is discussed in further detail above.

**References**

Hoffman, Matthew J., Xylar Asay-Davis, Stephen F. Price, Jeremy Fyke, and Mauro Perego. "Effect of subshelf melt variability on sea level rise contribution from Thwaites Glacier, Antarctica." *Journal of Geophysical Research: Earth Surface* 124, no. 12 (2019): 2798-2822.

Holland, Paul R. "The transient response of ice shelf melting to ocean change." *Journal of Physical Oceanography* 47, no. 8 (2017): 2101-2114.

Jonko, A., Urban, N.M. & Nadiga, B. Towards Bayesian hierarchical inference of equilibrium climate sensitivity from a combination of CMIP5 climate models and observational data. *Climatic Change* **149,** 247–260 (2018). https://doi.org/10.1007/s10584-018-2232-0

Rignot, Eric, Jérémie Mouginot, Bernd Scheuchl, Michiel Van Den Broeke, Melchior J. Van Wessem, and Mathieu Morlighem. "Four decades of Antarctic Ice Sheet mass balance from 1979–2017." *Proceedings of the National Academy of Sciences* 116, no. 4 (2019): 1095-1103.

Robel, Alexander A., Hélène Seroussi, and Gerard H. Roe. "Marine ice sheet instability amplifies and skews uncertainty in projections of future sea-level rise." *Proceedings of the National Academy of Sciences* 116, no. 30 (2019): 14887-14892.

References

Jenný Brynjarsdóttir and Anthony O'Hagan 2014 Inverse Problems 30 114007 (website: http://www.rdcep.org/research- projects/climate-emulator, source code: https://github.com/RDCEP/climate_emulator)

Chang, W., Haran, M., Applegate, P.J., and Pollard, D. (2016) Improving ice sheet model calibration using paleoclimate and modern data, the Annals of Applied Statistics, 10 (4), 2274-2302

Sacks, Jerome; Welch, William J.; Mitchell, Toby J.; Wynn, Henry P. Design and Analysis of Computer Experiments. Statistical Science 4 (1989), no. 4, 409-423. doi:10.1214/ss/1177012413. https://projecteuclid.org/euclid.ss/1177012413

Castruccio, Stefano, David J. McInerney, Michael L. Stein, Feifei Liu Crouch, Robert L. Jacob, and Elisabeth J. Moyer. "Statistical emulation of climate model projections based on precomputed GCM runs." *Journal of Climate* 27, no. 5 (2014): 1829-1844.

Salter, M. J., Daniel B. Williamson, John Scinocca & Viatcheslav Kharin (2019) Uncertainty Quantification for Computer Models With Spatial Output Using Calibration-Optimal Bases, Journal of the American Statistical Association, 114:528, 1800-1814